# Family imprint reveals basin-wide patterns of Amazon forest embolism resistance

Amazon rainforests face intensifying water stress due to increases in vapour pressure deficit and changing hydrological regimes. Embolism resistance ($\Psi_{50}$) is a critical metric of tree survival under drought conditions, it is defined as a plant's capacity to resist disruption of xylem water flow due to air bubble formation from water stress. However, measurements of $\Psi_{50}$ are only available for a limited number of Amazon locations and species. Conversely, data on forest taxonomic composition are abundant across Amazonia, and if $\Psi_{50}$ is conserved phylogenetically, these data could provide a way to scale-up drought resistance patterns. Here we evaluate $\Psi_{50}$ measurements across non-flooded Amazonian tree taxa and reveal a moderate phylogenetic signal, with phylogenetic conservatism evident at the family-level. Notably, Fabaceae is amongst the most embolism-resistant tree families in Amazonia. Leveraging the phylogenetic signal we use species composition and tree size data from 448 forest plots across Amazonia to produce a macroecological assessment of Amazonian vulnerability to embolism. The resulting estimate spatial pattern reveals that forests in the Brazilian and Guiana Shield regions, where Fabaceae abundance is high, show strong resistance to embolism. In contrast, tree communities in Western Amazonia appear more vulnerable to embolism, suggesting a reduced capacity to withstand future drought conditions.

The Amazon region is home to the largest and most diverse tropical forest in the world, and plays an important role in planetary biogeochemical cycles. Recent findings have documented substantial changes in non-flooded Amazonian forests (*terra-firme*), including floristic and functional composition[1–3], structure, and dynamics[4,5], potentially associated with ongoing changes in climate and atmospheric composition. In recent decades, the Amazon has been subjected to a sequence of large-scale drought events (1998, 2005, 2010, 2015–16, 2023 and 2024)[6–12], as well as a continued warming of 0.6°–0.7 °C since 1950[13], exposing plants to higher water stress due to increased vapour pressure deficit (VPD). Climate model projections also suggest that the severity of drought effects on forests will continue to increase and that temperatures will likely rise to levels without historical analogues[14,15]. Together, these climatic changes are expected to exacerbate water stress in Amazon rainforests.

Xylem embolism resistance is a key structural trait that determines the ability of plants to tolerate water stress. This is because water stress is associated with increasingly negative xylem water potentials, which may result in the formation of water vapour/air bubbles (emboli) in the xylem and compromise water transport to the canopy[16]. Typically, embolism resistance is quantified as the xylem water potential at which a tree's hydraulic conductivity declines to 50% of its maximum value[17] ($\Psi_{50}$), with a more negative $\Psi_{50}$ implying higher embolism resistance. In Amazonia, embolism resistance has been shown to explain transpiration and canopy conductance responses to extreme drought[18,19], as well as patterns of species distributions[20,21] and differential mortality patterns under imposed drought[22].

In recent years, advances have been made in understanding how embolism resistance varies locally[18,19,22–29] and along basin-wide pre-

✉ e-mail: tavares.juliav@gmail.com

cipitation gradients[21]. These studies reveal that embolism resistance varies in response to broader climatological differences[21] but also local topographical variation in water availability (e.g., refs. 20,30). Still, a key challenge remains in that embolism resistance is a complex and time-demanding trait to measure, thus far only measured at a small number of locations across Amazonia and therefore making inference of large-scale patterns difficult. Inventory data on tree species composition and dominance, by contrast, are now widely available across Amazonia (*e.g.*, the RAINFOR network encompasses 600 forest plots[31]). If embolism resistance has a phylogenetic signal, then the widespread availability of forest inventory data could enable the use of phylogenetic imputation to scale up trait observations. This approach allows missing values to be estimated based on the assumption that closely related species tend to share similar traits[32].

Embolism resistance has indeed been shown to be phylogenetically constrained at a global scale[33,34], indicating that a species' resistance to embolism is shaped by its ancestral lineage. Yet, Amazonian tree species are still very poorly represented in such analyses, which span global gradients of water availability and a vast amount of evolutionary clades[33,34], potentially leading to a stronger phylogenetic signal than what can be expected for regional-scale analyses. Efforts to determine the phylogenetic conservatism of embolism resistance in Amazonian trees have been made by studies in Central Amazonia at two sites[20,25]. These studies considered 28 and 16 congeneric species found on valley/plateau or flooded/non-flooded Amazonian habitats, respectively, and found no phylogenetic signal in embolism resistance. Rather, $\Psi_{50}$ values of species growing in flooded areas with ample water availability were significantly larger than those of congeneric species found in non-flooded areas. This suggests repeated convergent evolution of resistance to embolism based on topo-hydrological position, likely driven by the strong environmental pressure that these environments impose[25].

While apparently inconsistent with global studies, existing results for Amazonia are based on hydraulic traits datasets covering only a few taxa (e.g., ref. 25 considered six families and eight genera). Given that the Amazon contains more than 6000 known tree species belonging to 803 genera and more than one hundred families[35,36], a wider evaluation is essential to fully determine whether or not a phylogenetic signal in embolism-and drought-resistance exists in the Amazon tree flora.

Here we make use of the pan-Amazonian hydraulic trait database[21,37], encompassing data for 129 species, 88 genera, 36 families and 14 orders, combined with the most up-to-date Amazonian molecular-genus-level phylogeny[38] to show the extent to which embolism resistance across non-flooded (*terra-firme*) Amazonian tree taxa is phylogenetically conserved. We then integrate data on hydraulic traits and floristics from lowland non-flooded forests across the entire Amazon Basin[31] to create the most comprehensive macro-ecological assessment of Amazonian xylem embolism resistance to date. Our dataset includes the most important Amazonian tree families (e.g., Fabaceae, Moraceae, Lecythidaceae and Lauraceae), both in terms of number of stems[35,36,39] and aboveground biomass and wood productivity[40], allowing us to evaluate the relative importance of family vs. genus-level controls on $\Psi_{50}$ variation and to investigate differences in $\Psi_{50}$ across the major Amazonian families.

## Results

### Evidence of phylogenetic control on embolism resistance of Amazonian trees

We found evidence of a moderately strong phylogenetic signal (PS) for embolism resistance of Amazonian tree genera (Fig. 1A, Tab. 1), as demonstrated by a value of Blomberg's $k$ of 0.44 ($p = 0.02$) for $\Psi_{50}$ across the entire dataset. This value is at the top-end of reported $k$ values for tropical forest tree traits; by comparison, wood density across Amazonian taxa was found to show a phylogenetic signal with a Blomberg's k value of 0.30[41]. This is, however, still lower than the value

expected under a Brownian motion model of evolution[42]. The observed phylogenetic signal for embolism resistance remained similar when also accounting for environmental variation (*i.e.* excluding forests with long dry season length) (SI Fig. 1, Tab. 1). Nested analysis of variance on a sub-sample of our dataset ($n = 10$ families, 25 genera, 68 species), further revealed that family ($F = 2.411$, df = 9, $p = 0.003$) is a stronger predictor of $\Psi_{50}$ than genus ($F = 1.501$, df = 15, $p = 0.171$) or species ($F = 3.389$, df = 39, $p = 0.261$), implying that the phylogenetic control of hydraulic properties in Amazonian trees is not a statistical artefact. The sub-sampling was performed to ensure adequate replication within groups and to meet the test requirements.

### Fabaceae have particularly high embolism resistance

Phylogenetic randomisation analysis[43] revealed that the Myrtales and Fabales orders were particularly important in influencing the magnitude of the phylogenetic signal within our dataset, both having marked resistance to embolism (Fig. 1B). Among the families, Fabaceae and Rubiaceae stand out as particularly embolism-resistant. An analysis of variance (ANOVA) constrained to 12 widely-occurring Amazon tree families - those present in at least three sites (SI Figs. 2, 3), revealed significant differences in embolism resistance across major Amazon tree families (ANOVA and Tukey HSD post hoc: $F = 2.45$, df = 6, $p = 0.05$), with Fabaceae standing out as having, on average, particularly low values of $\Psi_{50}$ (i.e., more resistant, Fig. 2A). In contrast, Myristicaceae and Euphorbiaceae had lower embolism resistance (less negative $\Psi_{50}$), while Moraceae had intermediate resistance. Notably, Fabaceae consistently ranked among the most embolism-resistant families across the entire dataset, regardless of sampling density across sites and species (SI Fig. 4).

Fabaceae are the most abundant and ecologically dominant plant family in Amazonia[35,36,39,40], as well as being broadly distributed and dominant across South and Central American and African tropical rainforests[44]. Given their overarching ecological importance, we then tested whether Fabaceae, on average, are more emboli-resistant than non-Fabaceae within Amazonia. We found that Fabaceae (mean $\Psi_{50}$: $-2.6 \pm$ std 1.0) have a significantly more resistant xylem than non-Fabaceae (mean $\Psi_{50}$: $-1.8 \pm$ std 0.8) across our entire dataset (Fig. 2B, $W = 938$, $p = 0.00015$), with the difference being especially marked in the Amazonian forests with an intermediate dry season length that occupy most of the Amazon basin (SI Fig. 5, $W = 228$, $p = 0.0007$). A similar pattern was found for ever-wet aseasonal forests (SI Fig. 5, $W = 32$, $p = 0.035$), although the sample size of Fabaceae in this region was much smaller than in other regions. In long dry season length forests where species are more adapted to water stress, Fabaceae and non-Fabaceae have equally resistant xylems (SI Fig. 5, $W = 83$, $p = 0.922$). Overall, Fabaceae span a broad range of $\Psi_{50}$ values, reflecting the environmental and taxonomic diversity of the family (SI Fig. 4). Yet, despite this wide variation, within each site across the precipitation gradient, Fabaceae tend to consistently rank among the more embolism-resistant taxa, occupying the resistant end of the hydraulic spectrum across Amazonia forest plots (SI Fig. 6). Analysis of a broader global dataset[45] also yielded similar results, with Fabaceae showing greater xylem resistance when compared to other families, and being among the top most emboli-resistant families across tropical seasonal forests, tropical rainforests and temperate seasonal forests (SI Fig. 7).

### Embolism resistance of Amazonian tree communities

Based on the phylogenetic signal observed in our database, we used inventory data from 448 permanent plots across Amazonia (from the RAINFOR forest data network, curated by ForestPlots.net[31]) to impute $\Psi_{50}$ values to forest plots for which community-level $\Psi_{50}$ measurements were not available[21]. We gap-filled missing data using the approach of ref. 1, assigning values from the next available taxonomic level (see "methods"). After gap-filling, we calculated the community

 

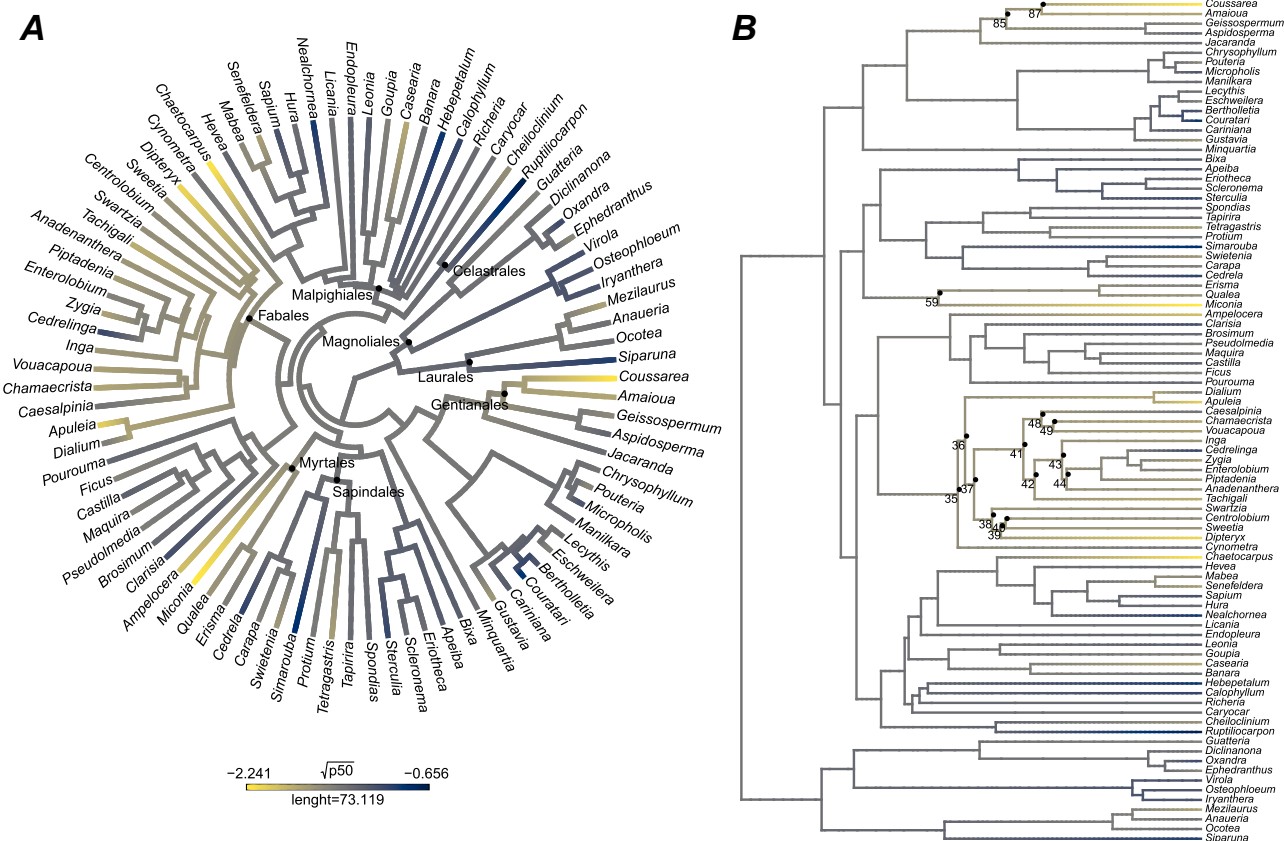

**Fig. 1 | Phylogenetic patterns of embolism resistance across amazonian tree genera. A** Phylogenetic tree of Amazonian tree genera with branches coloured by transformed absolute mean $\Psi_{50}$ values per genus, $n = 87$ genera, from yellow (most drought-resistant) to blue (least drought-resistant). **B** Randomisation analysis[43]: map of phylogenetic signal at genus-level for $\Psi_{50}$. The circles show individual nodes that have more resistant embolism values than expected randomly, at a 0.05 level of significance. Phylogenetic structure is based on ref. 38.

weighted mean (CWM) of each plot to capture the central tendency of embolism resistance at each site, weighting each species' trait value by its relative dominance (species basal area / total plot basal area). As these plots include information on species composition, abundance, and the size of each tree, we were able to estimate community-level embolism resistance across Amazonia, accounting for the relative local dominance of each taxon. Our estimates reveal a broad macro-ecological pattern of basal-area-weighted mean $\Psi_{50}$ (Fig. 3) where forests in the Brazilian and Guiana shields have tree communities with particularly high embolism resistance. Conversely, Western Amazon forests generally have communities with the least resistant xylem in the Amazon Basin. These patterns were shown to be consistent across two spatial analysis methods (interpolation and spatial-environmental clustering) and regardless of interpolation parameters used (SI Fig. 8, SI Figs. 13–15, and see "methods").

We also tested whether distance to water table depth (obtained from ref. 46, a key factor driving local water availability), would be related to the pan-Amazonian variation of embolism resistance (SI Fig. 9C). In contrast to what has been found across Amazonia and Cerrado biomes[30], our results indicate that variation in water table depth did not contribute to explain patterns of $\Psi_{50}$ distribution at the basin-wide level in Amazonia (statistical results are shown in the figure caption). However, the role of distance to water table on tree drought resistance manifests mainly at a local scale[20,47,48] and may not be accurately represented at the levels of generalisation and uncertainty associated with coarse water table depth estimations. These limitations could potentially mask the combined effects of water table depth and regional precipitation gradients that have been shown in previous studies.

## Discussion

Our results show that, on a pan-Amazonian scale, there is clear evidence of a phylogenetic signal for embolism resistance. This indicates that closely-related Amazon tree taxa have $\Psi_{50}$ values which are more similar than would be expected by chance, and thus evolutionary history plays an important role in determining embolism resistance. However, the signal ($k = 0.44$) is substantially weaker than expected under a Brownian motion (BM) model of evolution, indicating that other processes are also important[49]. Values of $k$ can vary continuously from zero to infinity, where a $k = 0$ indicates no phylogenetic signal and $k >= 1$ a strong phylogenetic signal, meaning that traits of close taxa tend to be more similar than traits of distant taxa[42]. Convergent evolution in distantly related lineages or divergent selection among closely related taxa both tend to reduce the levels of phylogenetic signal compared to a BM model[42], thus suggesting that these selection processes may have also played a role in the evolution of embolism resistance variation among Amazon trees[20,25,48]. Additionally, the family-level signal we found suggests a deeper-rooted evolutionary control on $\Psi_{50}$, which would not be detected by ref. 25. possibly due to the limited taxonomic scope surveyed.

Our finding that tropical Fabaceae show substantially higher resistance to embolism than non-Fabaceae taxa is particularly noteworthy given the status of Fabaceae as the most dominant angiosperm tree family in the tropical americas[44]. Several factors have been proposed to explain the success of Fabaceae across Amazonia. In primary forests, Fabaceae may be more successful in slow-turnover environments (low mortality of individuals and/or few natural forest disturbances), showing higher shade tolerance than other families in poor soils due to both their ability to fix nitrogen and their higher seed

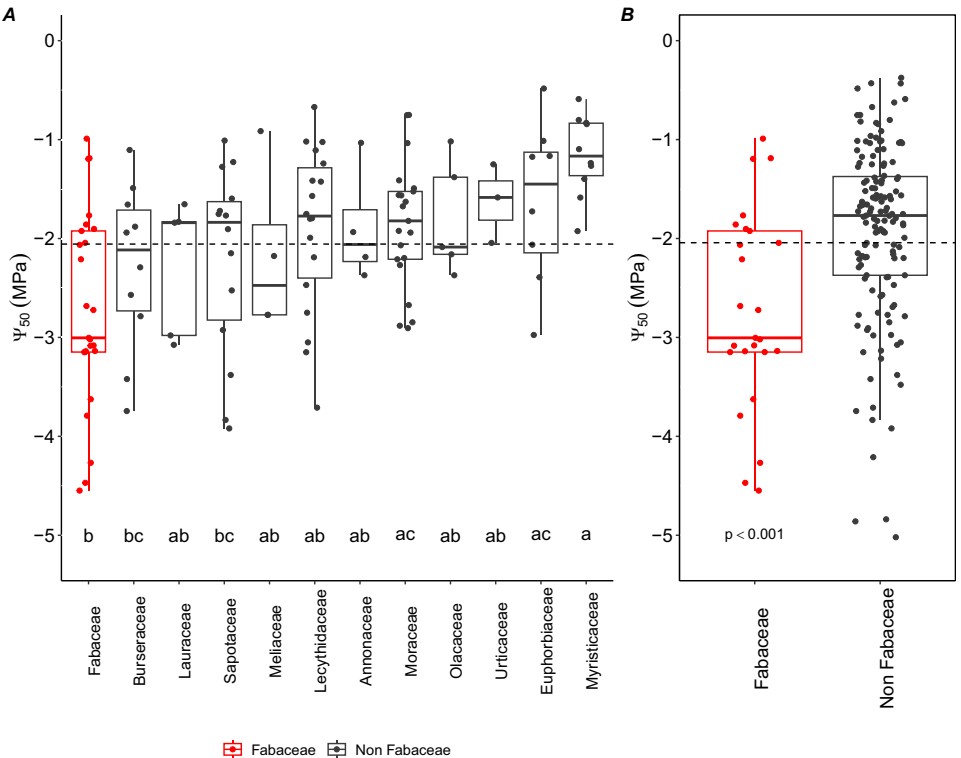

**Fig. 2 | Variation in embolism resistance across amazonian tree families.**
**A** Embolism resistance ($\Psi_{50}$) variation across widely-occurring Amazon tree families (SI Figs. 2, 3). Boxplots of $\Psi_{50}$ variation across families show the 25th percentile, median and 75th percentile. Vertical bars show the interquartile range × 1.5, and data points beyond these bars are potential outliers. Differences among families were tested using a one-way ANOVA ($F = 3.399$, df = 11, $p = 0.000421$) and Tukey HSD *post hoc* at a 0.05 significance level were performed. Significant pairwise differences are displayed on the figure as distinct letters. Points represent species value per site. Dashed horizontal lines show the overall mean $\Psi_{50}$ across widely occurring families. **B** Comparison of $\Psi_{50}$ between Fabaceae and non-Fabaceae across our entire dataset. Differences were tested using the Wilcoxon rank sum test with continuity correction ($W = 938$, $p = 0.0001545$). Boxplots indicate median, 25th and 75th percentiles; whiskers represent 1.5 × IQR; points beyond whiskers represent outliers. The dashed vertical line shows the mean $\Psi_{50}$ across the full pan-Amazonian dataset. SI Fig. 4 shows the full distribution of all sampled families in the dataset, regardless of sampling density across sites and species.

mass[39]. In secondary forests, the combination of Fabaceae's ability to fix nitrogen and their reduced leaflet size result in enhanced nutrient use efficiency and drought tolerance, explaining the ecological success of this group in regrowth areas[50,51]. To further explore whether embolism resistance in Fabaceae is associated with other functional traits, we compared leaf mass per area (LMA from refs. 21,37.) between Fabaceae and non-Fabaceae species and found no significant differences (SI Fig. 10). Within Fabaceae, we also tested whether specific subgroups (Caesalpinioideae, Dialioideae, and Papilionoideae), leaf architecture (bipinnate, pinnate, or unifoliate leaves), or nitrogen-fixation ability (data from ref. 51) explained variation in embolism resistance (SI Fig 10). However, our analysis indicates that embolism resistance is a general trait across Fabaceae, with no evidence that these factors drive variation in $\Psi_{50}$ values. While traits such as leaflet size, LMA, and nitrogen fixation likely contribute to Fabaceae's ecological success, our results suggest that their high xylem embolism resistance represents an additional, broadly shared trait that may help explain their remarkable abundance, dominance, diversity, and wide distribution across primary Amazonian forests. Indeed, Fabaceae are also markedly abundant in tropical dry forests[51,52], further supporting the hypothesis that their success is at least partly underpinned by high resistance to embolism.

Combining the phylogenetic signal found in our database with the extensive forest inventory information available across Amazonia allowed us to provide a comprehensive estimate of pan-Amazonian macroecological patterns in community-level vulnerability to embolism. Community-level embolism resistance has been shown to be highly related to background climate across Amazonian forests[21], and

our spatial analysis reflects both the distribution of precipitation in the basin and the strong effect of floristic composition (Fig. 3). The known precipitation gradient from northwestern to southern Amazon matched the vulnerability distribution in this direction, but we also found markedly high embolism resistance across the Brazilian and Guiana Shields, despite them having respectively low and high precipitation regimes[53]. The modelled high community-level embolism resistance in these areas seems instead to be at least partially driven by the high abundance of Fabaceae throughout both shields[39,54], which is also reflected in our community composition data. This relative dominance of Fabaceae had a strong positive effect on the resulting spatial pattern of community weighted mean of $\Psi_{50}$ (SI Fig. 9B, please see figure caption for statistical results), therefore indicating higher resistance to embolism. Our explicit spatial analysis also further indicates that tree communities in Western Amazonia seem to be characterised by less resistant xylems than those in Central and Eastern Amazonia (Fig. 3, SI Fig. 12 with statistical results in the caption)[21,55]. A similar macroecological pattern to ours has been observed for tree species turnover[56] across the basin, with a turnover gradient between western Amazonia and the Guiana and Brazilian Shields and a second gradient between northwest wet forests and southern drier forests. This further emphasises the role of phylogeny and species composition in determining embolism resistance in the Amazon basin.

Overall, our study shows that resistance to embolism is phylogenetically conserved for Amazonian tree species, and also suggests an apparent signature of processes such as convergent evolution and/or divergent selection. This indicates that phylogenetic constraints might provide boundaries for adaptation, but that adaptation to embolism

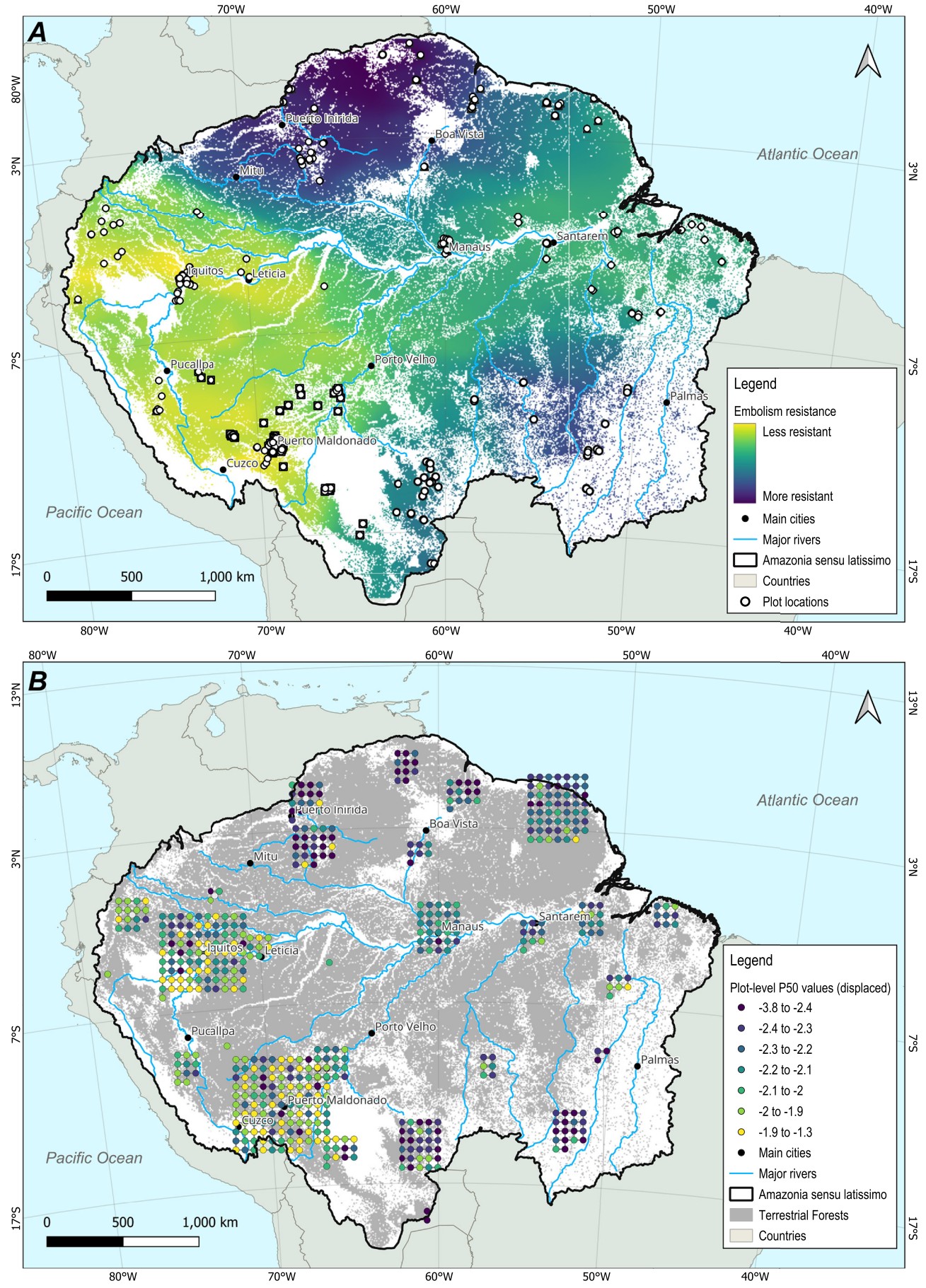

**Fig. 3 | Estimated basin-wide spatial variation of Amazonian vulnerability to embolism. A** Macroecological pattern of the community weighted mean value of $\Psi_{50}$. The pattern was created using Inverse Weighted Distance interpolation $\Psi_{50}$ of the estimated 448 upland moist forest plots distributed across Amazonia *sensu latissimo* (see "methods"). Our analyses exclude dry forests, flooded forests, and plots with elevation >1000 m above sea level, as well as those affected by direct human disturbance. **B** Estimated community weighted mean values for all 448 non- flooded forest plots across the basin. Each dot represents a forest plot and the colour shows its community-weighted mean $\Psi_{50}$. Due to the spatial scale of the map, plot locations in panel B are displaced (jittered) to remove overlap and improve visualisation; exact plot locations are shown in SI Fig. 8. On both panels, we masked out areas of Amazonia with very different environments, notably flooded forests, white sand forests and deforested areas, to emphasise our focus on upland (*terra-firme*) moist forests (see "methods" for details).

resistance does occur over evolutionary time. The extent to which species can potentially further adapt under ongoing and future climate change is still uncertain, as they may not be able to evolve fast enough to cope with new conditions[57,58]. Alternatively, under intensifying droughts and a warming climate, the species community composition of Amazonian forests may be expected to shift towards more embolism-resistant taxa. While findings for West African forests support this expectation, where multi-decadal long-term drought has been found to substantially alter community composition and lead to a moderate increase in Fabaceae abundance[3], recent results for Amazonian forests show that community composition does not seem to track climatic change[59], raising concerns about potential lags in ecological and evolutionary responses. At the same time, it is important to acknowledge that $\Psi_{50}$ captures only one dimension of drought response. Embolism resistance reflects xylem-level tolerance to low water potentials, but whole-plant vulnerability to drought also depends on other factors such as rooting depth, stomatal regulation, and phenology[60], which differ across lineages. Consequently, more negative $\Psi_{50}$ values, such as those frequently observed in Fabaceae, may represent an evolutionary compensation for other traits (e.g., shallower rooting) rather than a direct indicator of superior drought tolerance *sensu lato*. These considerations highlight that our spatial patterns should be interpreted as reflecting embolism resistance distributions rather than definitive predictions of drought-induced mortality. With these caveats in mind, our pan-Amazonian spatial analysis highlights patterns and hypotheses emerging from the Amazon-wide hydraulic trait dataset[21]. The results suggest that intensified water stress could result in stronger floristic filtering in the western Amazon, where current communities are particularly vulnerable to embolism. As much of western Amazonia arboreal diversity is strongly mesophilic[61], continued increases in drought frequency may lead to more pronounced compositional shifts and, ultimately, loss of biodiversity.

## Methods

### Amazonian tree embolism resistance ($\Psi$50) dataset
In this study, we used species and genus mean values of $\Psi_{50}$ from 129 species, 88 genera, belonging to 36 families and 14 orders, obtained from the pan-Amazonian hydraulic traits dataset[21], which integrates newly collected samples from Western and Southern Amazonia together with published data from Central–Eastern Amazonia[19,24,29]. The dataset is deposited as a ForestPlots.net data package, which can be accessed using the ref. 37. In this dataset, embolism resistance was characterised by constructing xylem vulnerability curves for 129 species among 11 sites encompassing effectively the entire Amazon climatological gradient, incorporating forests with: long dry season length, intermediate dry season length, and ever-wet aseasonal forests[21]. Xylem vulnerability curves were constructed for each species at each site by fitting a curve based on pooled data from individuals of the same species, quantifying xylem embolism formation as a function of branch dehydration[17]. To do so, ref. 21 used the pneumatic method[62], which consists of measuring the air discharge from terminal branch ends to assess embolism formation, and the bench dehydration technique to induce water stress in a given branch[62–65].

### Phylogenetic signal
We used the most up-to-date Amazon tree genus-level phylogeny[38] to explore the existence of phylogenetic signal (PS) in $\Psi_{50}$ across different Amazonian genera. To quantify the strength of PS, we used Blomberg's $k$ values, which provide a measure of the strength of phylogenetic signal based on comparing the observed variance for a given trait against the variance that would be expected under a Brownian motion (BM) model of trait evolution[42]. While '$k$' values close to zero imply evolutionary independence (random trait distribution among the branches of the phylogenetic tree), values close to one denote phylogenetic non-independence, indicating high trait similarity among closely related clades. The significance ($p$-value) of $k$ was estimated through a randomisation exercise in which the tips of the phylogenetic tree were randomised 1000 times and the resulting distribution of $k$ values was compared to the observed value of $k$. We defined $k$ values to be significant if they fell outside the 2.5% − 97.5% percentile range of the simulated distribution. Blomberg's $k$ test has been shown to be able to detect PS on trees with at least 20 observations and is thus appropriate for the size of our dataset[42]. We further tested the sensitivity of the observed phylogenetic signal (PS) by repeating the analyses on a restricted subset of the dataset. As the dataset spanned dry-adapted long dry season length forests as well as core Amazon forests, we re-ran the analyses excluding these long dry season forests to verify that observed PS patterns were not driven simply by differential sampling of genera across different climate regimes. Data were transformed to meet the normality assumptions of the test (Table 1). Due to the skewed distribution, a cube root transformation was applied when a square root transformation was not sufficient to normalise the data. This choice was guided by visual inspection of histograms and the Shapiro-Wilk normality statistical test. To test which specific taxonomic groups may strongly contribute to PS, we employed the randomisation approach of ref. 43. In brief, the approach consists of first estimating the ancestral value for each node in the phylogeny using ancestral state reconstruction and then randomising the tips of the phylogeny 1000 times to generate a random distribution of ancestral nodes, which are compared to the observed reconstructed node value.

Complementarily, to better understand the variation in $\Psi_{50}$ explained by different taxonomic levels (species, genus, family), we performed a nested analysis of variance (ANOVA) [family/genus/species]. To ensure the minimum replication within groups and meet the requirements of the test, this analysis was limited to genera for which we had data from more than one species and families with at least two genera or genera/families with the same species/genera sampled in more than 1 site. Data were square root transformed to meet the normal distribution criteria of the test. The reduced dataset in total accounted for 10 families, 25 genera, and 68 species.

### Embolism resistance variation across families
To evaluate embolism resistance variation across widely occurring families in our dataset, we conducted one-way ANOVA followed by Tukey's Honestly Significant Difference (HSD) tests. In this analysis, we selected only families that occurred in at least 3 sites along a wide mean MCWD (Maximum Climatological Water Deficit) gradient to strike a balance between including a broad representation of

**Table 1 | Summary table of phylogenetic signal (PS) for embolism resistance of Amazonian trees**

| Dataset | Transformation | Embolism resistance (MPa) | PS (k) | p value | N genera |
|---|---|---|---|---|---|
| Full dataset | Square root (Sqrt) | ψ50 | 0.44 | 0.02 | 87 |
| Long DSL forest excluded | Cube root (Cbrt) | ψ50 | 0.46 | 0.03 | 67 |

The strength and significance level of the phylogenetic signal for $\Psi_{50}$ as measured by Blomberg's $k$ are shown for the full dataset and for sensitivity analysis, whereby long dry season length (DSL) forests were excluded to account for environmental variation. Normality assumptions were met using data transformations: log-transformation for the full dataset and cube-root transformation when long-DSL sites were excluded due to increased right-skewness.
*DSL = dry season length

Amazonian tree diversity while maintaining statistical robustness in family-level comparisons across the precipitation gradient (Fig. 2A, SI Fig. 2).

Given the importance of Fabaceae both in this dataset (SI Figs. 3, 4) and across Amazonian forests more generally we also investigated whether there were differences in embolism resistance between Fabaceae and non-Fabaceae by performing a Wilcoxon rank sum test with continuity correction (Fig. 2B). We did this across the entire dataset but also for specific forest types (long dry season length, intermediate dry season length, and ever-wet aseasonal forests), following ref. 21 (SI Fig. 5). We used the global dataset from ref. 45 to investigate the distribution of Fabaceae embolism resistance in other systems. We focused our analyses on plant families represented by at least two individuals per biome, excluding the temperate rainforests due to limited Fabaceae representation. Winteraceae was excluded due to its lack of vessels (SI Fig. 7).

### Geographic patterns of Amazon embolism resistance

Based on the phylogenetic signal observed in our database, we computed the community-weighted mean $\Psi_{50}$ for a further 448 Pan-Amazonian plots from the RAINFOR inventory network[31]. Individual tree data for each plot were obtained from the ForestPlots.net database[66,67]. We selected single censuses with the best available level of taxonomic identification at the species level, for each structurally mature lowland tropical forest plot within the Amazonia *sensu latissimo* definition[68], therefore excluding dry forests, flooded forests, and plots with elevation >1000 m above sea level, as well as those affected by direct human disturbance. As the results of the nested ANOVA indicated a dominant family control on embolism resistance, we restricted the mapping exercise to plots that shared at least 60% of their dicotyledonous arboreal family composition, in basal area terms, with our $\Psi_{50}$ database.

We gap-filled missing data following the methods of ref. 1, whereby we used values for the next available taxonomic level. For example, for species for which we had not measured $\Psi_{50}$, we used genus-level means when available and family-level means when genus-level data were not available. When family-level data were not available, we used the plot mean value. After this procedure, we then calculated the community weighted mean (CWM) to describe the central tendency of tree community trait values at each site. The CWM50 was calculated by weighting the trait value of each species by its relative dominance (species basal area divided by plot total basal area) in each study plot. To test the gap-filling procedure, we replaced the original species values by family-mean values (simulating gap-filling) for the plots on which $\Psi_{50}$ were directly sampled[21] (SI Fig. 11B linear model: $p < 0.0001$; $R^2 = 0.78$) and for the surrounding plots with similar species composition[21] (SI Fig. 11A linear model: $p < 0.0001$; $R^2 = 0.72$) and compared the resulting CWM $\Psi_{50}$ values.

Once the plot dataset was gap-filled, we used two combined procedures to estimate macroecological/geographic patterns of $\Psi_{50}$ across Amazonia. First, we performed a geographically constrained clustering of the gap-filled data points, using the *ClustGeo* package[69] of R programming language version 4.4.2[70]. This method implements a Ward-like hierarchical clustering algorithm that includes spatial and environmental constraints. The algorithm takes two dissimilarity matrices, *DO* representing environmental distances (i.e., 'feature space') and *D1* representing geographic differences (i.e., 'constraint space'), and a mixing parameter *alpha* varying between 0 and 1. The criterion minimised at each stage is a convex combination of the homogeneity criterion calculated from *DO* and the homogeneity criterion calculated from *D1*, with the relative weight of each given by *alpha*. The value of *alpha* is optimised by graphical analysis of the relative proportion of explained inertia associated with *DO* and *D1*, so that the ideal *alpha value* increases spatial contiguity without deteriorating the quality of the solution based on the feature space.

We evaluated the effect of *alpha* values ranging from 0 to 1 at 0.1 intervals, for a total number of clusters ranging from 2 to 8, using plot-level $\Psi_{50}$, Mean Cumulative Water Deficit (MCWD) and annual and monthly Water Table Depth (WTD) as our feature space. Regardless of the number of clusters tested, the optimal *alpha* value was consistently 0.3. We then visually compared clustering results from 2 to 8 clusters, and decided as ideal the largest number of clusters that did not produce clusters with excessive spatial overlap between clusters, nor clusters with too few data points. Our final choice of clustering parameters was then *alpha* = 0.3 and $k = 5$ clusters, with the resulting clusters and corresponding average $\Psi_{50}$ values for each cluster shown on SI Fig. 8.

We then used the interpolation method of Inverse Distance Weighting (IDW)[71] to better visualise this basin-wide geographic pattern as a continuous surface, using the *gstat* package[72] running on R version 4.4.2[70]. IDW results are mainly contingent on two parameters, the inverse distance power coefficient (*idp*), which determines the strength of the decay of the weighting with respect to distance, and the maximum number of neighbour points to be considered when calculating a new interpolated value (*nmax*). To optimise the choice of these parameters, we used a Leave One-Out Cross Validation (LOOCV) strategy where several parameter combinations were tested by interpolating $N-1$ observations in the dataset, and then calculating the error between the interpolated value and the actual value of the left-out observation. This process is repeated $N$ times to generate a set of $N$ error values for each parameter combination. The parameter combination that minimised the Root Mean Squared Error (RMSE) was selected as optimal. After a three-step parameter search procedure, we converged on the best IDW parametrisation of *idp* = 0.3 and *nmax* = 32, and used these parameters to generate interpolated surfaces for all five folds of the cross validation (see below), averaged these five results to produce a final interpolation map over a raster grid with a cell size of $10 \times 10$ km, using the WGS-84 grid and the Equal Earth Americas projection (EPSG 8858). To reduce linear artefacts resulting from the uneven spatial distribution of the original plots, we smoothed each resulting interpolation using a $9 \times 9$ mean convolution filter prior to averaging.

To estimate accuracy, we performed a spatially-constrained $k$-fold cross-validation using the five spatial clusters as data folds to quantify fine-grained spatial uncertainty. For each *fold*, interpolation was performed for $N-1$ folds (training data), while the remaining fold (testing data) was used to calculate the Root Mean Squared Error and the cross-validation based Variance Explained (VEcv) uncertainty metrics[73]. Overall, quantitative agreement between field values and interpolated values had an RMSE of 0.24 MPa, with the RMSE value for each fold

varying from 0.19 MPa to 0.32 MPa (SI Fig. 13). We can see on the figure that most of the mismatch between observed and interpolated values comes from the smoothing inherent to the IDW interpolation method. For this reason, the calculated Variance Explained (VEcv) was only 6.7%. We further tested the adequacy of IDW as a visualisation tool by assessing the overall agreement of the IDW interpolated trends with the average values of the spatial clusters generated. First, we calculated the mean $\Psi_{50}$ value for each spatial cluster using the source point data (*point averages*). We then delineated the minimum convex hull enclosing all data points in each cluster, and used these polygons to extract all $\Psi_{50}$ pixel values from the interpolated raster, which were averaged to provide a second mean $\Psi_{50}$ value (*interpolated averages*). We then compared the agreement between field-based and interpolation-based averages for all five clusters, showing that both had a very good agreement (SI Fig. 14).

Finally, to test the robustness of our generalisation, we performed Multivariate Environmental Similarity Surface (MESS) analysis[74], using the *dismo* R package, version 1.3.16[75]. This analysis tests how well each location in the prediction (interlation) space matches the environmental conditions of the source data (field points). Values between 0 and 100 indicate that the location is within the min-max range of all environmental variables, with 50 indicating that all environmental variables are right in the centre of their min-max ranges. Environmental suitability is calculated individually for each environmental variable at each location, and the final MESS value for the locations is given as the worst similarity score across all variables. MESS values below zero indicate that at least one of the environmental variables at that location exceeds the environmental envelope of the source data, and therefore, spatial predictions should be taken with caution. We calculated MESS statistics for our interpolated surface using the Minimum Cumulative Water Deficit and Water Table Depth raster layers as environmental variables, and extracted their corresponding values for each of the 448 field plots as reference values for MESS calculation. The analysis indicated that over 93% of the study area (Amazonia *latissimo sensu*) had MESS values within the range of 0 to 100, with areas below zero corresponding to isolated pixels, the highlands Andean region and a few regions in the lowland western Amazonia (SI Fig. 15). When plotting the data, we have added for context a terrestrial forest mask, produced by combining the Amazon wetlands mask[76], Brazilian land cover types from MapBiomas[77], and the latest deforestation layer from the Brazilian PRODES monitoring programme[78]. Main rivers were obtained from the World Bank Data Catalogue[79].

Taken as a whole, the multiple approaches to accuracy analysis emphasise that our interpolation and clustering approaches do succeed at showing the broad, basin-wide macroecological patterns of vulnerability to embolism that are already present in the estimated plot-level values. Importantly, however, neither of these procedures produces pixel-level accurate spatial predictions of $\Psi_{50}$ across Amazonia - and thus we strongly discourage attempts to use the IDW results as a 'data layer' for further analysis.

### Reporting summary
Further information on research design is available in the Nature Portfolio Reporting Summary linked to this article.

## Data availability
The embolism resistance dataset used in this study is available through the pan-Amazonian hydraulic traits dataset[21], which is deposited as a ForestPlots.net data package: https://doi.org/10.5521/forestplots.net/2023_1. This dataset integrates newly collected samples from Western and Southern Amazonia together with published data from Central–Eastern Amazonia[19,24,29]. The phylogenetic tree used in our analyses is accessible in the ref. 38. Community-weighted mean $\Psi_{50}$ estimates for Pan-Amazonian plots used in this study are available

through the ForestPlots.net data package[80]: https://doi.org/10.5521/forestplots.net/2025_5

## Code availability
Code to reproduce the main analyses and figures is available as part of the ForestPlots.net data package associated with this study[80]: https://doi.org/10.5521/forestplots.net/2025_5

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

## Acknowledgements

This paper is an outcome of J.V.T.'s doctoral thesis, which was sponsored by Coordenação de Aperfeiçoamento de Pessoal de Nível Superior - Brasil (CAPES, finance code 001, GDE 99999.001293/2015-00). J.V.T. was previously supported by the NERC-funded ARBOLES project (NE/S011811/1) and is currently supported by a Birgitta Sintring Foundation (Stipend S2023-0009, Sweden) to J.V.T. and by the Swedish Research Council Vetenskapsrådet (grant no. 2019-03758 to R.M.). Data collection was largely funded by the UK Natural Environment Research Council (NERC) project TREMOR (NE/N004655/1) to D.G., E.G. and O.L.P., with further funds from CAPES - Brasil (GDE 99999.001293/2015-00) to J.V.T. and a University of Leeds Climate Research Bursary Fund to J.V.T. D.G., E.G. and O.L.P. acknowledge further support from a NERC-funded consortium award (ARBOLES, NE/S011811/1). E.G., O.L.P and D.G. acknowledge support from NERC-funded BIORED grant (NE/N012542/1). O.P. acknowledges support from an ERC Advanced Grant and a Royal Society Wolfson Research Merit Award. R.S.O. was supported by a CNPq productivity scholarship, the São Paulo Research Foundation (FAPESP-Microsoft 11/52072-0) and the US Department of Energy, project GoAmazon (FAPESP 2013/50531-2). M.M. acknowledges support from MINECO FUN2FUN (CGL2013-46808-R) and DRESS (CGL2017-89149-C2-

1-R). P.M. acknowledges support from the Royal Society Wolfson Fellowship RSWF_211008 and NERC (NE/W006308/1). E.N.H.C. was supported by NERC Knowledge Exchange Fellowship (grant ref no. NE/V018760/2). K.R. thanks the Aarhus University Research Foundation grant AUFF-E-2023-7-3 to Hanna Tuomisto. K.Y. was funded in part by the Consejo Nacional de Ciencia, Tecnología e Innovación Tecnológica (CONCYTEC) and the Programa Nacional de Investigación Científica y Estudios Avanzados (PROCIENCIA) within the framework of the E033-2023-01-BM "Alianzas Interinstitucionales para Programas de Doctorado" contest, grant number (PE501084299-2023). This paper is facilitated by the RAINFOR plot network and its long-term forest records curated at ForestPlots.net. As well as investigators and field leaders included here, we gratefully acknowledge the efforts of several hundred additional botanists, technicians and field assistants who contributed to the installation, measurement and identification of trees across South American forests. RAINFOR and ForestPlots.net have been supported by numerous people and grants since their inception. For their contributions to developing the RAINFOR network and antecedents, we are indebted to our late colleagues Elisbán Armas, Terry Erwin, Thomas Lovejoy, Alwyn Gentry, Sandra Patiño, Antonio Peña Cruz, David Neill and Jean-Pierre Veillon. For financial support we thank the European Research Council (ERC Advanced Grant 291585—'T-FORCES'), the Gordon and Betty Moore Foundation (#1656 'RAINFOR', and 'MonANPeru'), the European Union's Fifth, Sixth and Seventh Framework Programme (EVK2-CT-1999-00023—'CARBONSINK-LBA', 283080—'GEOCARBON', 282664—'AMAZALERT'), the Natural Environment Research Council (NE/D005590/1—'TROBIT', NE/F005806/1—'AMAZONICA', NE/X014347/1—'AMSINK'), several NERC Urgency and New Investigators Grants, the NERC/State of São Paulo Research Foundation (FAPESP) consortium grants 'BIO-RED' (NE/N012542/1), 'ECOFOR' (NE/K016431/1, 2012/51872-5, 2012/51509-8), 'ARBOLES' (NE/S011811/1, FAPESP 2018/15001-6), Brazilian National Research Council (PELD/CNPq 403710/2012-0), the Royal Society (University Research Fellowships and Global Challenges Award ICA/R1/180100 - 'FORAMA'), the National Geographic Society (PFA-21-PP029), US National Science Foundation (DEB 1754647) and Colombia's Colciencias. This manuscript is an output of ForestPlots.net Research Project 19, " Hydraulic properties of Amazonian trees: spatial variation and consequences for vulnerability to drought". ForestPlots.net is a meta-network and cyber-initiative developed at the University of Leeds to unite permanent plot records and support tropical forest scientists. We thank A. Levesley, K. Melgaço Ladvocat and G. Pickavance for ForestPlots.net management. We acknowledge the contributions of the ForestPlots.net Collaboration and Data Request Committee (B.S.M., E.N.H.C., O.L.P., T.R.B., B. Sonké, C. Ewango, J. Muledi, S.L.L., L. Qie) for facilitating this project and associated data management. Data curation, partner support, and the development of ForestPlots.net have been funded by grants including NE/B503384/1, NE/N012542/1 - 'BIO-RED', ERC Advanced Grant 291585 - 'T-FORCES', NE/F005806/1 - 'AMAZONICA', NE/N004655/1 - 'TREMOR', NE/X014347/1 - AMSINK', NERC New Investigators Awards, the Gordon and Betty Moore Foundation ('RAINFOR', 'MonANPeru'), ERC Starter Grant 758873 -'TreeMort', EU Framework 6, and a Leverhulme Trust Research Fellowship.

## Author contributions

J.V.T., D.G., E.G., and T.S.F.S. designed the study with inputs from R.S.O., M.M., and O.L.P. Data analyses were done by J.V.T. and T.S.F.S. with inputs from D.G., E.G., F.C.S., O.L.P., M.M., R.S.O., T.R.B., and A.E.-M. The manuscript was written by J.V.T. with main inputs from D.G., E.G., T.S.F.S., F.C.S., R.S.O., P.B., L.R., P.M., M.M., O.P., C.S.-M., K.G.D., R.M., E.N.H.C., M.D., J.C., R.B., and I.O.M., J.V.T., C.S.-M., F.C.D., M.G., L.P., M.A., M.J.M.Z., C.A.S.Y., F.M.P.-M., H.J., M.C.S., P.B., J.A.R.S., and R.T.O. collected hydraulic traits data. Basin-wide forest species composition data were collected, managed or funded by F.C.S., B.S.M., B.H.M.J., Y.M., I.O.M., L.R., P.M., A.C.L.C., A.E.-M., E.Á.-D., M.N.A., E.A.O., A.A., L.A., A.A.-M., L.Ar., G.A., J.G.B., D.B., R.B., C.C., J.L.C., R.S., W.C., J.C., J.C.,

D.C.D., G.D., M.D., A.D., S.F., T.F., G.F.L., B.H., L.H., N.H., E.N.H.C., E.J.-R., M.K., S.L., W.L., S.L., A.S.L., A.M.-M., P.M., P.N.V., D.N., W.P., A.P.G., G.P.-M., M.C.P.-M., N.P., R.R., A.P., M.R.-M., H.R.-A., S.C.R., K.R., R.P.S., J.S., R.S., A.R.R., M.S., H.t.S., J.T., L.V.G., R.V.M., I.V., E.V.T., V.A.V., O.W., K.Y., R.M., T.R.B., O.L.P., and D.G. All authors critically revised the manuscript.

## Funding

## Competing interests

The authors declare no competing interests.

## Additional information

Julia Valentim Tavares [1,2] ✉, Emanuel Gloor[2], Thiago S. F. Silva[3], Rafael S. Oliveira[4], Fernanda Coelho de Souza[5], Caroline Signori-Müller[4,6], Francisco Carvalho Diniz[2], Luciano Pereira[7], Martin Acosta[8], Martin Gilpin[2], Manuel J. Marca Zevallos[9,10], Carlos A. Salas Yupayccana[9], Flor M. Perez-Mullisaca[9], Halina Jancoski[11], Marina Corrêa Scalon[12], Beatriz Schwantes Marimon[11], Ben Hur Marimon Junior[11], Yadvinder Malhi[13], Imma Oliveras Menor[13,14], Lucy Rowland[6], Patrick Meir[15,16], Paulo Bittencourt[17], Antonio Carlos Lola da Costa[18,19], João Antônio R. Santos[20], Renata Teixeira de Oliveira[20], Adriane Esquivel-Muelbert[21,22], Esteban Álvarez-Dávila[23], Miguel N. Alexiades[24], Edmar Almeida de Oliveira[11], Ana Andrade[25], Luiz Aragão[26], Alejandro Araujo-Murakami[27,28], Luzmila Arroyo[28], Gerardo Aymard[29], Jorcely G. Barroso[30], Damien Bonal[31], Roel Brienen[2], Carlos Céron[32], José Luís Camargo[25], Richarlly Silva[33], Wendeson Castro[20], Jérôme Chave[34], James Comiskey[35,36], Douglas C. Daly[37], Geraldine Derroire[38,39,40], Mathias Disney[41], Aurelie Dourdain[38], Sophie Fauset[42], Ted Feldpausch[6], Gerardo Flores Llampazo[43], Bruno Hérault[39,44], Lionel Hernández[45], Niro Higuchi[46], Eurídice N. Honorio Coronado[47], Eliana Jimenez-Rojas[48], Michelle Kalamandeen[49], Susan Laurance[50], William Laurance[50], Simon Lewis[2,41], Antonio S. Lima[19], Abel Monteagudo-Mendoza[9,51], Paulo Morandi[11], Percy NúñezVargas[9], David Neill[52,76], Walter Palacios[53], Alexander Parada Gutierrez[29], Guido Pardo-Molina[54], Maria Cristina Peñuela-Mora[55], Nigel Pitman[56], Rocio Rojas[51], Adriana Prieto[57], Maxime Réjou-Méchain[14], Hirma Ramírez-Angulo[58], Sabina Cerruto Ribeiro[59], Kalle Ruokolainen[60,61], Rafael P. Salomão[19,62], Julio Serrano[46], Rodrigo Sierra[63], Ademir R. Ruschel[64], Marcos Silveira[20], Hans ter Steege[65,66], John Terborgh[67,68], Luis Valenzuela Gamarra[51], Rodolfo Vásquez Martinez[51], Ima Vieira[19], Emilio Vilanova Torre[69], Vincent A. Vos[54], Ophelia Wang[70], Kenneth Young[71,72], Robert Muscarella[1], Kyle G. Dexter[15,73], Timothy R. Baker[2], Oliver L. Phillips[2], Maurizio Mencuccini[74,75] & David Galbraith[2]

[1]Plant Ecology and Evolution, Evolutionary Biology Centre, Uppsala University, Uppsala, Sweden. [2]School of Geography, University of Leeds, Leeds, UK. [3]Biological and Environmental Sciences, Faculty of Natural Sciences, University of Stirling, Stirling, UK. [4]Departamento de Biologia Vegetal, Instituto de Biologia, Universidade Estadual de Campinas (UNICAMP), Campinas, Brazil. [5]BeZero Carbon, London, UK. [6]Department of Geography, Faculty of Environment, Science and Economy, University of Exeter, Exeter, UK. [7]Institute of Botany, Ulm University, Ulm, Germany. [8]Centro Avançado de Pesquisa-Ação da Conservação e Recuperação Ecossistêmica da Amazônia (CAPACREAM), Campinas, Brazil. [9]Universidad Nacional de San Antonio Abad del Cusco, Cusco, Peru. [10]Instituto de la Naturaleza, Tierra y Energía, Pontificia Universidad Católica del Perú, Lima, Peru. [11]Programa de Pós-Graduação em Ecologia e Conservação, Universidade do Estado de Mato Grosso (UNEMAT), Nova Xavantina, MT, Brazil. [12]Programa de Pós-Graduação em Ecologia e Conservação, Universidade Federal do Paraná, Curitiba, Brazil. [13]Environmental Change Institute, School of Geography and the Environment, University of Oxford, Oxford, UK. [14]AMAP, Univ. Montpellier, IRD, CNRS, CIRAD, INRAE, 34000 Montpellier, France. [15]School of Geosciences, University of Edinburgh, Edinburgh, UK. [16]Research School of Biology, Australian National University, Canberra, ACT, Australia. [17]School of Earth and Environmental Sciences, Cardiff University, Cardiff, UK. [18]Instituto de Geociências, Faculdade de Meteorologia, Universidade Federal do Pará, Belém, Brazil. [19]Museu Paraense Emílio Goeldi, Belém, Pará, Brazil. [20]Laboratório de Botânica e Ecologia Vegetal, Universidade Federal do Acre, Rio Branco, Brazil. [21]Department of Plant Sciences, University of Cambridge, Cambridge, UK. [22]School of Geography, University of Birmingham, Birmingham, UK. [23]Escuela de Ciencias Agrícolas, Pecuarias y del Medio

Ambiente, National Open University and Distance, Bogotá, Colombia. [24]Environmental and Rural Science, University of New England, Armidale, New South Wales, Australia. [25]Biological Dynamics of Forest Fragment Project, INPA and STRI, Manaus, Brazil. [26]National Institute for Space Research (INPE), São José dos Campos-SP, Brazil. [27]Museo de Historia Natural Noel Kempff Mercado, Santa Cruz, Bolivia. [28]Universidad Autonoma Gabriel Rene Moreno, Santa Cruz, Bolivia. [29]UNELLEZ-Guanare, Programa de Ciencias del Agro y el Mar, Herbario Universitario (PORT), Mesa de Cavacas, Venezuela; Jardín Botánico de Bogotá José Celestino Mutis, Cl. 63 #68-95, Bogotá, DC, Colombia. [30]Centro Multidisciplinar, Universidade Federal do Acre, Cruzeiro do Sul, Acre, Brazil. [31]Université de Lorraine, AgroParisTech, INRAE, UMR Silva, 54000 Nancy, France. [32]Herbario Alfredo Paredes (QAP), Universidad Central del Ecuador, Quito, Ecuador. [33]Instituto Federal de Educação, Ciência e Tecnologia do Acre, Campus Baixada do Sol, Rio Branco, Brazil. [34]Laboratoire Evolution et Diversité Biologique (EDB) CNRS/UPS, Toulouse, France. [35]Inventory and Monitoring Program, National Park Service, Fredericksburg, VA, USA. [36]Smithsonian Institution, Washington, DC, USA. [37]The New York Botanical Garden, Southern Boulevard, New York, NY, USA. [38]Cirad, UMR EcoFoG (AgroParistech, CNRS, INRAE, Université des Antilles, Université de la Guyane), Campus Agronomique, Kourou, French Guiana. [39]CIRAD, UPR Forêts et Sociétés, Montpellier, France. [40]University of Brasilia, Department of Forestry, Federal District, Brasilia, Brazil. [41]Department of Geography, University College London, London, UK. [42]School of Geography, Earth and Environmental Science, University of Plymouth, Plymouth, UK. [43]Instituto de Investigaciones de la Amazonia Peruana, Iquitos, Peru. [44]Forêts et Sociétés, Univ Montpellier, CIRAD, Montpellier, France. [45]Centro de Investigaciones Ecológicas de Guayana, Universidad Nacional Experimental de Guayana, Estado Bolívar, Venezuela. [46]Instituto Nacional de Pesquisas da Amazônia, Manaus, Brazil. [47]Royal Botanic Gardens, Kew, London, UK. [48]Instituto IMANI, Universidad Nacional de Colombia, Leticia, Colombia. [49]Unique land use GmbH, Schnewlinstraße 10, 79098, Freiburg im Breisgau, Breisgau, Germany. [50]Centre for Tropical Environmental and Sustainability Science and College of Science and Engineering, James Cook University, Cairns, Queensland, Australia. [51]Jardín Botánico de Missouri, Oxapampa, Peru. [52]Facultad de Ingeniería Ambiental, Universidad Estatal Amazónica, Puyo, Ecuador. [53]Universidad Tecnica del Norte, Herbario Nacional del Ecuador, Quito, Ecuador. [54]Instituto de Investigaciones Forestales de la Amazonía, Universidad Autónoma del Beni José Ballivián, Riberalta, Bolivia. [55]Universidad Regional Amazónica IKIAM, Tena, Ecuador. [56]Field Museum of Natural History, Chicago, IL, USA. [57]Instituto de Ciencias Naturales, Universidad Nacional de Colombia, Bogotá, Colombia. [58]Instituto de Investigaciones para el Desarrollo Forestal, Universidad de Los Andes, Mérida, Venezuela. [59]Centro de Ciências Biológicas e da Natureza, Universidade Federal do Acre, Rio Branco, Brazil. [60]Biodiversity Unit of the University of Turku, Turku, Finland. [61]Department of Biology, Aarhus University, Aarhus, Denmark. [62]Universidade Federal Rural da Amazônia—UFRA/CAPES, Belém, Brazil. [63]Geoinformática & Sistemas (GeoIS), Quito, Ecuador. [64]Embrapa Florestas, Embrapa, Colombo, Paraná, Brazil. [65]Naturalis Biodiversity Center, Leiden, the Netherlands. [66]Quantitative Biodiversity Dynamics, Department of Biology, Utrecht University, Utrecht, the Netherlands. [67]Florida Museum of Natural History and Department of Biology, University of Florida, Gainesville, FL 91, USA. [68]School of Science and Engineering, James Cook University, Cairns, Queensland, Australia. [69]Verra, Washington, DC, USA. [70]School of Earth Sciences and Environmental Sustainability, Northern Arizona University, Flagstaff, AZ, USA. [71]Department of Geography and the Environment, University of Texas, Austin, USA. [72]Facultad de Ciencias Biológicas, Programa de Doctorado, Universidad Nacional Mayor de San Marcos Lima, Lima, Peru. [73]Botanic Garden Edinburgh, Edinburgh, UK. [74]CREAF, Campus UAB, Cerdanyola del Vallés, Barcelona, Spain. [75]ICREA, Barcelona, Spain. [76]Deceased: David Neill ✉e-mail: tavares.juliav@gmail.com

