## [Transparent Peer Review file · Nature Communications]

Family imprint reveals basin-wide patterns of Amazon forest embolism resistance

Corresponding Author: Dr Julia Tavares

Version 0:

Reviewer comments:

Reviewer #1

(Remarks to the Author)

This is an interesting study that tests the degree of phylogenetic signal in xylem vulnerability to cavitation among a sample of Amazonian tree species. The results suggest a significant phylogenetic signal with Fabaceae exhibiting more negative values as a group.

Major issues- The study is a little too Amazon-focused. Certainly this is an important system and might be used to model rainforest evolutionary patterns, but these data need to be presented in the context of other systems. Many important questions raised by this study were not considered. Such as:

1. How do these P50 values compare with values from other clades and communities?
2. P50 and "drought sensitivity" are assumed to be the same thing. If this is the case then a more detailed consideration of the combined effect of phylogeny and climate on the distribution of P50 might be expected.
3. The range of P50 overall seems very small compared with some other studies of communities (uncited work by Blackman, Brodribb Powers and others). This makes the community of the Amazon trees seem rather convergent in general, but his component is missed by failing to consider other systems.
4. Are Fabaceae "tougher" than other families in other systems?
5. How does the P50 of Amazonian Fabaceae compare with other data?

These are a few obvious and important questions that highlight the problem with being too focused on Amazonia. I suggest a significant broadening of perspective would greatly benefit the impact of the work.

Reviewer #2

(Remarks to the Author)

Review Report for Nature Communications

J. V. Tavares, et al.

"Family imprint reveals basin-wide patterns of Amazon forest embolism resistance"

Degradation of Amazonian tropical forests due to rapid climate change and land use pressure is an essential concern of the Earth system. Whether and how tree species diversity plays the role of tropical forest resilience remain challenging questions. By this paper, authors paid attention to the branch xylem resistance to embolism among angiosperm tree taxa across Amazonian forests, using the recent database of tree species' Ψ_{50} values compiled by Tavares et al. (2023) in relation to phylogeny and biogeographic distributions based on the region-wide database of tree inventory plots. Authors highlighted that genera of Fabaceae in particular exhibit high xylem embolism resistance in terms of low Ψ_{50} values than genera of other families, and low- Ψ_{50} species tended to be distributed in drier forest types and climatic subregions. Together with the previous papers (Oliveira et al. 2019; Tavares et al. 2023) indicating that Ψ_{50} as a reasonable indicator of drought resistance variation across tree species and forest types on the Amazonian basin, this paper offers interesting information for understanding the role of phylogenetic diversity and history in forest functioning and predicting the tree community change in Amazonian forests under anthropogenic pressure.

Here I would pose some points to be considered for easier understanding of wide-ranged readers of this paper.

General points:

(1) On non-Fabaceous taxa

The Ψ_{50} dataset indicates a wide variation across taxa. Some families and genera are characterized by high or low Ψ_{50} . On line 283-284, you demonstrate the two orders, Fabales and Myrtales, “significantly” and “markedly” distinct from others based on the phylogenetic linkage analysis. Fig. 1A also indicates genera of other than these two orders show high (and low) resistance as well. Figures 1 and 2 also demonstrate there is large cross-taxa overlap in Ψ_{50} . Under the title of “family imprint”, it is better not (only) mentioning two characteristic APG orders, but indicate typical families in Ψ_{50} distribution, e.g. Rubiaceae by genus *Coussarea*. ‘Opposite’ high Ψ_{50} taxa such as Myristicaceae would be worth mentioned, because they would be more sensitive to ongoing rapid climate change.

(2) On other traits of Fabaceae

As is briefly discussed (line 386-387), Fabaceae species are characterized by symbiotic nitrogen fixation and compound leaves. Authors discuss that small leaflet size reflecting high nitrogen content would underlie high drought resistance of this family. Meantime, it has been suggested that leaves with high nitrogen content tend to have low leaf mass per area (LMA), short leaf lifespan, thus quick turnover of leaves (e.g., Reich, 2014, *J. Ecol.*). Such leaf trait would contribute to drought resistance of legume tree species. In relation to quick leaf turnover, drought deciduousness is also key to drought resistance (e.g., Oliveira et al. 2021). These would not be necessarily linked to, but act complementary to xylem embolism resistance. It looks authors have collected dataset of LMA of those examined taxa (Tavares et al. 2023), I wonder it worth to compare LMA variation between Fabaceae and non-Fabaceae taxa.

Specific points:

line 152, line 170, ...: The term “terrestrial forest” (for Terra firme) sounds confusing. Better rephrase, e.g. “non-flooded forest”.

line 204-209: Only abundance or composition? May abundance change caused by tree deaths during severe drought events?

Table 1: It looks you made transformations for “absolute” Ψ_{50} values, because water potential is always negative. Explain clearly. Also provide reasoning of cube root transformation rather than square root of “absolute” Ψ_{50} for drier-ecotone excluded dataset (more skewed distribution of Ψ_{50}).

Fig. 1 (and SM Dig. 1): Fix inset color bar caption “sqrt(p50)”. Either $\sqrt{|\Psi_{50}|}$ or “square-root of absolute Ψ_{50} ”, and remove “-” (minus). For SM Fig. 1, “cube root of Ψ_{50} ” (not “absolute”).

line 291: Fabaceae is “broadly dominant across most of the world’s tropical forests” — to be rephrased. This statement is not the case for insular Southeast Asian tropical rainforests. There are some important species, but we do not observe any family-level dominance or abundance there, in contrast to continental American and African rainforests. I wonder your finding of high drought resistance of Fabaceae may provide a partial explanation of Fabaceae abundance in continental tropical climate experiencing drought periods.

Fig. 3B: In inset “P50” is “ Ψ_{50} ”.

line 391: “... and tropical forests more generally” better rephrase or remove as above.

line 436: Provide the elevation range of sites? (later it appears in line 502).

line 467-468: Define how to classify “dry ecotonal forests” from others. It appears unclear “ecotonal” in Results. Every time note “dry” or “drier”. Definition is also to be repeated in Results (e.g. line 268; Table 1).

line 469: Is “climate regimes” biome types? You can also incorporate climate measures (e.g. MCWD groups) instead of forest-biome types.

All figures are informative and collectively summarize well the paper.

SM Fig. 1: It would be better to provide comparable panel to Fig. 1B.

SM Figs 2 and 3 are interesting. Are some of these forest types “drier ecotonal forests”? If so, indicate them in figure legends.

Reviewer #3

(Remarks to the Author)

In this paper, the authors explore phylogenetic and spatial patterns in embolism resistance among tropical tree species. They find a moderate phylogenetic signal, with Fabaceae displaying an especially high embolism risk. The phylogenetically conserved signal allows them to produce a map of embolism risk across the Amazon basin, showing that Fabaceae-dominated systems likewise exhibit low risk.

This is a nicely written manuscript with a clear set of analyses and goal. I think it is timely given the focus on tree traits, and

the relative difficulty measuring and quantifying traits related to plant embolism risk and moisture regulation more broadly. However, from a methodological perspective, I think some of the decisions could be better justified, and some of the inferences are not fully supported by the current analyses.

First, the authors restricted their main analysis (e.g., Fig 2) to the widely occurring tree families. But doing so essentially limits this work to being only relevant to the 12 widely-occurring families. For example, what do the results look like if you use all families? Is Fabaceae just low risk among widely occurring families, or is it low even when the others are considered? If part of the goal of this work is to look across a wide number of taxa (as suggested by the authors, L 226-236), then this decision to restrict the data seems counterproductive, and the requirement of "six sites" used to determine widely occurring species seems arbitrary. It's also unclear if this subsetting is used throughout the results (e.g., L 294, L 378) or just for specific analyses. The choice to exclude families seems particularly relevant when discussing the relative difference of Fabaceae across forest types (L 294-300). These results suggest that Fabaceae is simply a uniquely widespread dry-adapted species. By excluding these dry-adapted specialists (who I presume are not widespread), then of course Fabaceae might look to be very dry-adapted among the generalists. A more careful treatment of this issue in the results (e.g., stratifying the ANOVA by forest type) could help shed light on this.

The authors also note that they exclude specific plots based on forest type and composition (L 503). But yet the resulting map is projected across the entire Amazon basin, without regard to forest type or composition. These decisions seem at odds with one another. Are the excluded plots regionally clustered or located on specific environmental conditions? If regions are excluded, then these should be masked from the extrapolation (or, more rigorously, something like Multivariate Environmental Similarity Surfaces should be used to prevent environmental extrapolation).

In line with this, the location of the forest plots is very highly clustered (Fig. 3), and there's no attempt to quantify the relative extent of spatial extrapolation (or environmental extrapolation). In such a setting, using inverse distance to interpolate the space seems suspect, and at a minimum, I would expect to see how predictive accuracy decreases as a function of decay distance. There's no reason to expect that regions 500 km from observed data points can be interpolated accurately, and the resulting map (Fig 3a) supports this, as it essentially just outlines the strong clusters with no ability to truly interpolate the space between.

To obtain a realistic goodness of fit for the mapping results, I'd like to see how these models perform using more rigorous spatially buffered leave-one-out CV or similar, using a kernel distance at which spatial autocorrelation drops to near zero (see Roberts et al. 2017, doi:10.1111/ecog.02881; Ploton et al 2020, doi: 10.1038/s41467-020-18321-y). I would also like to see the full raw cross-validation predictions for all LOO points, rather than the R2 of means (as in Fig. S5, which surely overestimates R2). I'd also be interested in seeing this cross-validation accuracy split up between the non-gap-filled points (L 508) and the gap-filled points, as I suspect the model performs worse if only worse due to the gap filling. I was also surprised not to see any spatial uncertainty, e.g., at a minimum using resampling or bootstrap approaches to quantify the role of sampling noise. As such, it's difficult to interpret the map with any confidence or make reliable inferences, apart from what is given in the raw plot data (e.g., Fig. 3B).

Lastly, given that the authors wish to explore the role of environmental drivers like water depth, is there a reason they did not use other mapping/modelling approaches that explicitly link embolism to covariates? The current approach of a simple linear model (Fig 8C) is likely confounded by a variety of underlying abiotic factors, such as precipitation and temperature regimes, soil types, etc. If the authors want to make robust inferences about water table depth, I would suggest a more structured approach to better control for confounders and interactions.

A few minor comments:

Supplementary Table 1 appears to be missing

What dataset did the water table depth come from? Is this Mattos et al?

The data description could have a bit more detail. Are all of the analyses and plots species-level averages? Did the original dataset contain multiple measurements within each species? If so, what the the level of intra vs. interspecific variation in the full dataset?

Version 1:

Reviewer comments:

Reviewer #1

(Remarks to the Author)

The MS first claims to show strong phylogenetic signal in xylem vulnerability, which is then used to impute P50 in forest plots where species were not measured, then interpolated across the landscape to create a "resistance to embolism?" map. This is an interesting method that creates a nice overview of possible landscape function, but the accuracy of this approach is highly questionable in my opinion.

A key finding is stated as the demonstration that Fabaceae are more embolism resistant than other families in Amazonia. Combined with the finding that "family ($F=2.74$, $p=0.02$) is a stronger predictor of ψ_{50} than genus ($F=1.88$, $p=0.07$) or species

($F=1.1$, $p=0.4$). This leads the reader to assume that much of the imputation is driven by family-level means. The problem with this approach is that the variation in P50 in Fabaceae spanned almost the entire range of all other taxa. When combined with the fact that the Fabaceae mean P50 was not significantly different from other families, one begins to see that the (potentially adaptive) variation that is likely to be captured by the predictions is extremely low.

Of the 3 outputs of this paper, none seem very robust.

1. That phylogeny has an influence on P50 is already known
2. That Fabaceae is a particularly tough does not seem particularly well supported considering the enormous diversity of P50 in the family (presumably correlated with the large diversity of the family).
3. That coarse scale mapping on P50 provides a useful insight into regional-scale "vulnerability" seems unlikely or at least demanding of a far more robust ground-truthing.

Thus, I feel that a much more cautious approach to the interpretation of these data is warranted. Having said this, I do believe there is promise in such an approach, so perhaps the manuscript simply requires a more circumspect rather than categorical viewpoint.

Reviewer #2

(Remarks to the Author)

Dear Authors,

I found authors revised the paper throughout in response to reviewers' comments and questions inclusive of my own. Added information in review responses added further information. Revised paper presents clear and new insights from the region-wide phylogenetic and geographic analysis focusing on stem embolism tolerance are worth to be published.

Reviewer 2

Reviewer #3

(Remarks to the Author)

I appreciate the clarification the authors have provided regarding the subsetting of data and the comparison of Fabaceae relative to other families. The additional analysis and clarification of the text address my key concerns in this area, providing convincing evidence that Fabaceae is not simply a uniquely widespread drought-tolerant specialist. I likewise appreciate the exclusion of specific forest types in the spatial projection, as well as the clustered cross-validation for overall model fit. Though, for this last point, what is the R^2 of observed vs. predicted rather than the RMSE? Depending on how the data are normalised, an RMSE of 0.24 could equate to an R^2 as low as $RMSE^2 = 0.05$ (see Li 2017 for a discussion on this point <https://doi.org/10.1371/journal.pone.0183250>).

With this in mind, there is still inadequate assessment of the uncertainty in the spatial projections. The authors state that they are unable to do traditional spatial uncertainty/extrapolation analyses because "the spatial distribution of the points is far from ideal to support quantitative modelling," and that, "with our clustered point distribution, no acceptable semivariance model could be fit." But this is exactly the issue -- if the data do not allow for appropriate statistical analysis, then how is the analysis justified? The authors state that, "our intent was not to quantitatively and accurately predict Ψ_{50} values spatially across the basin..." and that determining the extent of spatial extrapolation is, "beyond the scope of this paper..." Yet they are visually depicting these spatial patterns and referencing these patterns in the title, so one would hope that doing so "accurately" is an aim. These calculations (the MESS statistics, spatial uncertainty via bootstrapping, etc) are relatively easy to do with so few points, and would provide important context for understanding which regions are being estimated with higher/lower uncertainty.

The justification the authors provide for their approach is Mitchard et al. (2014). But in the decade since this paper was published, there has been growing consensus about the appropriate analysis of spatial patterns and what is necessary when producing spatial projections. It's certainly possible that the spatial projections are reasonable, but without estimates of uncertainty/extrapolation, it's equally likely that the patterns are heavily biased. The uncertainty/extrapolation analysis doesn't need to be incredibly laborious, but even some indication of which regions have higher/lower uncertainty would help. If the data do not allow for proper error assessment, or if the spatial projections have a small area of applicability or low goodness of fit, then I'm not sure that a spatial analysis is feasible given the current data.

Version 2:

Reviewer comments:

Reviewer #1

(Remarks to the Author)

Authors have responded to my previous comments with some justifications in their rebuttal, but without making any substantive changes to the text. My suggestion that there should be more consideration of potential weaknesses in the proposed arguments, has been ignored in the text. This is OK I suppose, but it does seem unfortunate to make bold statements about climate vulnerability, when it is possible that P50 patterns may more strongly reflect phylogenetic patterns than tolerance of drought, Consider the case if Fabaceae tend to be more shallow rooted than other families, their more

negative P50 will not correspond to a greater capacity to withstand drought, but rather a compensation for their shallow roots. Furthermore, if the presence of Fabaceae exerts a major influence on community "vulnerability", then this may not reflect a sensitivity of the community to rainfall shortage in the way that is being sold here. This is a theoretical scenario, but it is based on classical observations in the field, and by the general patterns observed in Choat et al 2012 (Nature).

Reviewer #3

(Remarks to the Author)

I appreciate the substantial effort the authors have invested in the additional spatial modelling and the clarity of analysis. I have no critiques with the overall question or analysis, nor do I have an issue with showing a spatial map per se, but I continue to remain skeptical about the robustness and usefulness of the current mapping approach.

As stated previously, my main concern is extrapolation beyond data support. Large parts of the central basin lack observations; thus the maps necessarily predict into unsampled regions. Despite the authors' reframing this as simply a "visualization", this is misleading---it is still an extrapolation into regions where there are no data---and needs to be evaluated as such. Ultimately, we need a way to distinguish reliable spatial projections from poor ones, or at least provide the readers with the necessary information for them to judge on their own.

Regarding predictive performance, I fully agree with the authors that spatial projections need not (and rarely do) deliver precise point estimates, but they must be approximately unbiased (i.e., track the 1:1 line) even if they have high point-to-point variance. That is, the discussion invoking Bergmann's rule appears to conflate inference about the mean trend with the width of prediction intervals; wide prediction intervals (low predictive accuracy at the point level) do not justify biased trends on aggregate. The new cross-validation (SM Fig. 13) suggests negligible predictive accuracy with substantial bias. And while SM Fig. 14 indicates good within-fit performance on aggregate, to justify extrapolating trends beyond clustered regions, this likewise needs to be conducted out-of-fit. However, even if the authors were to aggregate the clusters in SM Fig. 13 and plot obs vs. pred for each of the 5 folds, it appears that there would be no real signal, suggesting that even on aggregate, the Kriging approach does not yield approximately unbiased results.

The MESS analysis is a useful direction, but the variable choice needs clarification. Much of the text emphasizes Mean Cumulative Water Deficit and Water Table Depth, whereas the MESS uses "Maximum Cumulative Water Depth." If this is not a typo, please justify the choice and the exclusion of other plausible drivers (e.g., temperature, precipitation/BioClim, topography). Because MESS depends on the included variables, restricting to two requires a rationale; alternatively, show that these two suffice by demonstrating relatively unbiased spatial CV performance with them.

Given the current CV results, I remain hesitant about including Fig. 3 as a basin-wide product. It remains very possible that it is either misleading or essentially random outside of the sampled areas. If the editor prefers to treat the figure as a qualitative visualization rather than a quantitatively accurate projection, I'm happy to defer to that framing. But in that case, I think more caveats would be required, and perhaps adding a much coarser categorical classification (high/med/low), which presumably can be predicted with greater accuracy.

Responses to referees
Manuscript "Family imprint reveals basin-wide patterns of Amazon forest embolism
resistance"

Tavares et al.

Tracking Number **NCOMMS-24-52403**

Referees' comments:

Reviewer #1 (Remarks to the Author):

Comment 1.1. This is an interesting study that tests the degree of phylogenetic signal in xylem vulnerability to cavitation among a sample of Amazonian tree species. The results suggest a significant phylogenetic signal with Fabaceae exhibiting more negative values as a group. Major issues- The study is a little too Amazon-focussed. Certainly this is an important system and might be used to model rainforest evolutionary patterns, but these data need to be presented in the context of other systems. Many important questions raised by this study were not considered. Such as:

Response: We thank the reviewer for these comments on the novelty and significance of our manuscript. Following the referee's suggestion, we broadened the overall discussion of our paper, by adding information about Fabaceae in other systems (Choat et al., 2012) - please see responses to comments 1.4 and 1.5.

Reference: Choat, B., Jansen, S., Brodribb, T.J., Cochard, H., Delzon, S., Bhaskar, R., Bucci, S.J., Feild, T.S., Gleason, S.M., Hacke, U.G. and Jacobsen, A.L., 2012. Global convergence in the vulnerability of forests to drought. *Nature*, 491(7426), pp.752-755.

Comment 1.2. 1. How do these P50 values compare with values from other clades and communities?

Response: We believe this topic is answered in the responses to comments 1.4 and 1.5, on which we added global Ψ_{50} data from Choat et al (2012).

Comment 1.3. 2. P50 and "drought sensitivity" are assumed to be the same thing. If this is the case then a more detailed consideration of the combined effect of phylogeny and climate on the distribution of P50 might be expected.

Response: We agree with the referee that an improvement in the terminology was needed in the text. We therefore removed 'drought sensitivity' and just used the terminology 'resistance to embolism' throughout the text.

Comment 1.4. 3. The range of P50 overall seems very small compared with some other studies of communities (uncited work by Blackman, Brodribb Powers and others). This makes the community of the Amazon trees seem rather convergent in general, but his component is missed by failing to consider other systems.

Response: We respectfully disagree with the referee's assertion that the overall Ψ_{50} range in our study is significantly smaller than in previous studies. When placing our dataset in a broader context, we find that the range of Ψ_{50} values reported here - derived from Tavares et al. (2023) - broadly overlaps with values reported for tropical forests in Choat et al. (2012), please, see figure 1 below.

It is also important to highlight key methodological and ecological differences between our study and the cited works. Blackman, Brodribb, and Jordan (2012) and Powers et al. (2020) focused on leaf embolism resistance, which is expected to differ from branch xylem embolism resistance (Levionnois et al., 2020). Furthermore, their studies were conducted in tropical montane and dry forests, with some forests experiencing prolonged dry seasons (5–6 months dry season length - DSL) and receiving ~1600–1700 mm of mean annual precipitation. This contrasts with our sampling design, which incorporates aseasonal, intermediate, and long DSL lowland Amazonian rainforests. Given these climatic differences, the broader Ψ_{50} range in their dataset is expected. For comparison, Powers et al. (2020) reported Ψ_{50} values ranging from -1.55 to -6.52 MPa, while Blackman, Brodribb, and Jordan (2012), studying montane rainforest and dry sclerophyll forest, found values between -1.03 and -4.31 MPa. As shown in the figure below, our dataset reaches values as low as -5.1 MPa, demonstrating substantial overlap with previously reported ranges.

Reference: Levionnois, S., Ziegler, C., Jansen, S., Calvet, E., Coste, S., Stahl, C., Salmon, C., Delzon, S., Guichard, C. and Heuret, P., 2020. Vulnerability and hydraulic segmentations at the stem–leaf transition: coordination across Neotropical trees. *New Phytologist*, 228(2), pp.512-524.

Fig 1. Tree embolism resistance distribution across Amazonian terrestrial forests (Tavares et al., 2023) in grey compared with global vegetation (Choat et al., 2012) in red.

Comment 1.5. 4. Are Fabaceae “tougher” than other families in other systems?

Response: To address the referee’s question, we analyzed data from Choat et al. (2012), which evaluated xylem resistance to embolism (Ψ_{50}) across six biomes: boreal tundra, temperate rainforest, temperate seasonal forest, tropical rainforest, tropical seasonal forest, and woodland/shrubland. We focused on plant families represented by at least two individuals per biome, excluding temperate rainforests due to limited Fabaceae representation. Winteraceae was excluded due to its lack of vessels.

Fabaceae does not rank among the most resistant families (based on median Ψ_{50}) in mediterranean biomes (woodland/shrubland), which aligns with expectations given the hydraulic strategies of trees in these ecosystems. However, in tropical seasonal forests and tropical rainforests and even temperate seasonal forests, Fabaceae tend to exhibit greater xylem resistance compared to other families. The broader dataset confirms our findings that Fabaceae is among the top most emboli- resistant families.

Fig 2. (New SM Figure 6). Xylem embolism resistance (Ψ_{50}) across angiosperm families in different biomes, based on the global dataset from Choat et al. (2012). Panels show data for: (A) woodland/shrubland, (B) temperate seasonal forest, (C) tropical seasonal forest, and (D) tropical rainforest. Red points indicate Fabaceae species; black points represent species of other families. Dashed lines represent biome-wide means.

Comment 1.6. 5. How does the P50 of Amazonian Fabaceae compare with other data?

Response: Please see response to comments 1.4 and 1.5

Comment 1.7. These are a few obvious and important questions that highlight the problem with being too focused on Amazonia. I suggest a significant broadening of perspective would greatly benefit the impact of the work.

Response: We thank the referee for raising this point, which we believe indeed benefits the impact of the work. We attempted to incorporate this suggestion into the text (lines 310-314 and 525-529) and by adding SM Fig. 6.

Reviewer #2 (Remarks to the Author):

Review Report for Nature Communications J. V. Tavares, et al. "Family imprint reveals basin-wide patterns of Amazon forest embolism resistance"

Comment 2.1. Degradation of Amazonian tropical forests due to rapid climate change and land use pressure is an essential concern of the Earth system. Whether and how tree species diversity plays the role of tropical forest resilience remains a challenging question. By this paper, authors paid attention to the branch xylem resistance to embolism among angiosperm tree taxa across Amazonian forests, using the recent database of tree species' Ψ_{50} values compiled by Tavares et al. (2023) in relation to phylogeny and biogeographic distributions based on the region-wide database of tree inventory plots. Authors highlighted that genera of Fabaceae in particular exhibit high xylem embolism resistance in terms of low Ψ_{50} values than genera of other families, and low- Ψ_{50} species tended to be distributed in drier forest types and climatic subregions. Together with the previous papers (Oliveira et al. 2019; Tavares et al. 2023) indicating that Ψ_{50} as a reasonable indicator of drought resistance variation across tree species and forest types on the Amazonian basin, this paper offers interesting information for understanding the role of phylogenetic diversity and history in forest functioning and predicting the tree community change in Amazonian forests under anthropogenic pressure. Here I would pose some points to be considered for easier understanding of wide-range readers of this paper.

Response: We thank the referee for the comments and seek to address the referee's concerns in subsequent comments.

Comment 2.2. General points: (1) On non-Fabaceous taxa the Ψ_{50} dataset indicates a wide variation across taxa. Some families and genera are characterized by high or low Ψ_{50} . On line 283-284, you demonstrate the two orders, Fabales and Myrtales, "significantly" and "markedly" distinct from others based on the phylogenetic linkage analysis. Fig. 1A also indicates genera of other than these two orders show high (and low) resistance as well. Figures 1 and 2 also demonstrate there is large cross-taxa overlap in Ψ_{50} . Under the title of "family imprint", it is better not (only) mentioning two characteristic APG orders, but indicate typical families in Ψ_{50} distribution, e.g. Rubiaceae by genus Coussarea. 'Opposite' high Ψ_{50} taxa such as Myristicaceae would be worth mentioned, because they would be more sensitive to ongoing rapid climate change.

Response: Comment accepted. We added the referee's suggestion by mentioning more families explicitly on lines 288-297.

Comment 2.3. (2) On other traits of Fabaceae is briefly discussed (line 386-387), Fabaceae species are characterized by symbiotic nitrogen fixation and compound leaves. Authors discuss that small leaflet size reflecting high nitrogen content would underlie high drought resistance of this family. Meantime, it has been suggested that leaves with high nitrogen content tend to have low leaf mass per area (LMA), short leaf lifespan, thus quick turnover of leaves (e.g., Reich, 2014, *J. Ecol.*). Such leaf trait would contribute to drought resistance of legume tree species. In relation to quick leaf turnover, drought deciduousness is also key to drought resistance (e.g., Oliveira et al. 2021). These would not be necessarily linked to, but act complementary to xylem embolism resistance. It looks authors have collected dataset of LMA of those examined taxa (Tavares et al. 2023), I wonder it worth to compare LMA variation between Fabaceae and non-Fabaceae taxa.

Response: Following the referee's suggestion, we compared LMA variation between Fabaceae and non-Fabaceae species. Our results show no significant differences in LMA values between these groups (please see Figure 3 below, $W = 1229$, $p\text{-value} = 0.17$). Within Fabaceae, we further examined whether specific subgroups (Caesalpinioideae, Dialioideae, and Papilionoideae), leaf architecture (bipinnate, pinnate, unifoliate leaves), or nitrogen-fixation ability influenced the high embolism resistance observed in this family. However, our analysis indicates that embolism resistance is a general trait across Fabaceae, with no evidence that these factors drive variation in Ψ_{50} values. While these factors may contribute to Fabaceae's overall success, our results indicate that embolism resistance may be another layer contributing to the success of this Family and is independent of leaf traits such as LMA or nitrogen-fixation ability. We added this discussion into the text (lines 401-414) and added the figure below into the Supplementary Material (SM Fig. 9)

Fig 3. (New SM Figure 9). Fabaceae embolism resistance in variation across leaf traits, phylogenetic subgroups and nitrogen fixation ability. (A) Comparison of leaf mass per area (LMA) between Fabaceae (red) and non-Fabaceae (black) species across our entire dataset. (B–D) Variation in xylem embolism resistance (Ψ_{50}) within Fabaceae subgroups: (B) by phylogenetic subfamily (Caesalpinioideae, Dialioideae, Papilionoideae), (C) by leaf architecture (bipinnate, pinnate, unifoliolate), and (D) by nitrogen-fixation ability (yes/no; based on Gei et al., 2018). Red diamonds indicate group means; black dashed lines indicate the overall Fabaceae mean Ψ_{50} ; blue dashed lines represent the overall dataset mean.

Reference: Gei, M. et al. (2018) 'Legume abundance along successional and rainfall gradients in Neotropical forests', *Nature Ecology and Evolution*, 2(7), pp. 1104–1111.

Specific points:

Comment 2.4. line 152, line 170, ...: The term “terrestrial forest” (for Terra firme) sounds confusing. Better rephrase, e.g. “non-flooded forest”.

Response: We incorporate the referee’s suggestion by replacing “terrestrial forest” for “non-flooded forest” throughout the text.

Comment 2.5. line 204-209: Only abundance or composition? May abundance change caused by tree deaths during severe drought events?

Response: We understand that the referee has assumed that we are talking about 'abundance' in an ecological sense, instead, we used 'abundance' in a generic sense. To avoid confusion, we replaced 'abundance of forest inventory measurements' with 'widespread availability' of forest inventory data.

Comment 2.6. Table 1: It looks you made transformations for “absolute” Ψ_{50} values, because water potential is always negative. Explain clearly. Also provide reasoning of cube root transformation rather than square root of “absolute” Ψ_{50} for drier-ecotone excluded dataset (more skewed distribution of Ψ_{50}).

Response: We have clarified in the text (lines 279-281 and 496–499) that transformations were performed to the absolute values of Ψ_{50} to satisfy the test requirement of normality. Regarding the choice of cube root transformation rather than square root, when we excluded the drier ecotonal forests (long DSL) the distribution of Ψ_{50} values became more strongly right-skewed and a square root transformation was insufficient to achieve normality for statistical testing. We therefore applied a cube root transformation that better normalized the distribution. This choice was guided by visual inspection of histograms and Q-Q plots, as well as the Shapiro-Wilk normality statistical test.

Comment 2.7. Fig. 1 (and SM Dig. 1): Fix inset color bar caption “sqrt(p50)”. Either sqrt| Ψ_{50} | or “square-root of absolute Ψ_{50} ”, and remove “-” (minus). For SM Fig. 1, “cube root of Ψ_{50} ” (not “absolute”).

Response: We thank the referee for spotting this mistake, which is now fixed.

Comment 2.8. line 291: Fabaceae is “broadly dominant across most of the world’s tropical forests” — to be rephrased. This statement is not the case for insular Southeast Asian tropical rainforests. There are some important species, but we do not observe any family-level dominance or abundance there, in contrast to continental American and African rainforests. I wonder if your finding of high drought resistance of Fabaceae may provide a partial explanation of Fabaceae abundance in continental tropical climates experiencing drought periods.

Response: We rephrased the sentence as “Fabaceae are the most abundant and ecologically dominant plant family in Amazonia (Ter Steege *et al.*, 2013; Fauset *et al.*, 2015, Cardoso *et al.*, 2017), as well as being broadly distributed and dominant across South and Central American and African tropical rainforests (Gentry, 1988)”. (lines 300-301).

Comment 2.9. Fig. 3B: In inset “P50” is “ Ψ_{50} ”.

Response: We thank the referee for spotting this mistake, which is now fixed.

Comment 2.10. line 391: “... and tropical forests more generally” better rephrase or remove as above.

Response: Referee’s suggestion is now incorporated.

Comment 2.11. line 436: Provide the elevation range of sites? (later it appears in line 502).

Response: We think that the elevation range of the sites is better place in line 502 (first submitted version) now numbered as line 538 (current revised version), because it incorporates all the 448 forest plots used in the macroecological assessment, instead of mentioning this information for only the 11 sites from Tavares *et al.*, 2023, to which line 436 (first submitted version) is referring to.

Comment 2.12. line 467-468: Define how to classify “dry ecotonal forests” from others. It appears unclear “ecotonal” in Results. Every time note “dry” or “drier”. Definition is also to be repeated in Results (e.g. line 268; Table 1).

Response: Referee’s suggestion is now incorporated. We replaced “ecotonal” from the results and kept the same terminology used in Tavares et al., (2023) through the text: long dry season length, intermediate dry season length, and ever-wet aseasonal forests.

Comment 2.13. line 469: Is “climate regimes” biome types? You can also incorporate climate measures (e.g. MCWD groups) instead of forest-biome types.

Response: Referee’s suggestion is now incorporated, please see response to the comment above.

Comment 2.14. All figures are informative and collectively summarize the paper well.

Response: We thank the referee for the comment.

Comment 2.15. SM Fig. 1: It would be better to provide a comparable panel to Fig. 1B.

Response: Referee’s suggestion is now incorporated.

Comment 2.16. SM Figs 2 and 3 are interesting. Are some of these forest types “drier ecotonal forests”? If so, indicate them in figure legends.

Response: Referee’s suggestion is now incorporated. We now indicate in the legend which are the long DSL, intermediate DSL and ever-wet aseasonal forests.

Reviewer #3 (Remarks to the Author):

Comment 3.1. In this paper, the authors explore phylogenetic and spatial patterns in embolism resistance among tropical tree species. They find a moderate phylogenetic signal, with Fabaceae displaying an especially high embolism risk. The phylogenetically conserved signal allows them to produce a map of embolism risk across the Amazon basin, showing that Fabaceae-dominated systems likewise exhibit low risk.

This is a nicely written manuscript with a clear set of analyses and goal. I think it is timely given the focus on tree traits, and the relative difficulty measuring and quantifying traits related to plant embolism risk and moisture regulation more broadly. However, from a methodological perspective, I think some of the decisions could be better justified, and some of the inferences are not fully supported by the current analyses.

Response: We thank the referee for these comments on the novelty and significance of our manuscript and seek to address the referee’s concerns in subsequent comments.

Comment 3.2. First, the authors restricted their main analysis (e.g., Fig 2) to the widely occurring tree families. But doing so essentially limits this work to being only relevant to the 12 widely-occurring families. For example, what do the results look like if you use all families? Is Fabaceae just low risk among widely occurring families, or is it low even when the others are considered?

Response: We restricted the main analysis in Fig. 2A, specifically subsetting families that occurred in 3 or more sites in our dataset, totalling 12 families. However, we agree with the reviewer that it is also valuable to examine the full distribution of all sampled families. We have

now included this extended comparison in SM Fig. 4 (also shown below), which presents the Ψ_{50} values across all families in the dataset, regardless of sampling density across site and species. This allows readers to assess the relative position of Fabaceae within the broader phylogenetic context.

Finally, we would like to emphasize that our Fabaceae vs. non-Fabaceae comparisons (e.g., in Fig. 2B) used data from all non-Fabaceae taxa in the dataset—not only those from the 12 most widely occurring families. The restriction to 12 families applies only to the left panel of Fig. 2A, which was intended to highlight family-level differences across the full Amazonian precipitation gradient. We made the subsetting rationale and when it is applied more clear to the readers in lines (289-290, 301-307 and 513-529)

Fig. 4 (New SM Figure 4). Embolism resistance variation across all sampled families in the dataset, regardless of sampling density across site and species. Dashed horizontal lines show mean trait value across families. Boxplots show the 25th percentile, median and 75th percentile. Vertical bars show the interquartile range $\times 1.5$ and data points beyond these bars are potential outliers. The dashed vertical line shows the mean Ψ_{50} across the pan-Amazonian dataset.

Comment 3.3 If part of the goal of this work is to look across a wide number of taxa (as suggested by the authors, L 226-236), then this decision to restrict the data seems counterproductive, and the requirement of "six sites" used to determine widely occurring species seems arbitrary.

Response: We thank the reviewer for raising this point and we revised the text accordingly (lines 289-290, 301-307 and 513-529). Through the referee's question we realised that our text contained a mistake and also it was not clear enough to the reader. Our decision to restrict the main family-level comparison (Fig. 2A) to widely occurring families was made to ensure that differences in embolism resistance reflected intrinsic taxonomic or functional variation rather than environmental filtering linked to narrow geographic or climatic distributions. For this, we selected species that occurred in at least three sites (not six - this was a mistake in the text), please see SM Fig. 2. We have chosen this to strike a balance between including a broad representation of Amazonian tree diversity while maintaining statistical robustness in family-level comparisons across the precipitation gradient.

Comment 3.4. It's also unclear if this subsetting is used throughout the results (e.g., L 294, L 378) or just for specific analyses. The choice to exclude families seems particularly relevant when discussing the relative difference of Fabaceae across forest types (L 294-300). These results suggest that Fabaceae is simply a uniquely widespread dry-adapted species. By excluding these dry-adapted specialists (who I presume are not widespread), then of course Fabaceae might look to be very dry-adapted among the generalists. A more careful treatment of this issue in the results (e.g., stratifying the ANOVA by forest type) could help shed light on this.

Response: We appreciate the referee's thoughtful comment and acknowledge that the distinction between analyses based on the full dataset versus the subset of widely occurring families was not always made clear in the original text. To improve clarity, we have now revised the Results section to explicitly state, at each relevant point (e.g., Lines 288-297 and figure captions), whether the full dataset or a subset was used, along with the rationale behind each choice. Specifically, for the analysis presented in Lines 294–300 (submitted version), we used the entire dataset, not only the subset of widely occurring families. The subset of 12 families was used exclusively for the analysis shown in Fig. 2A (as discussed in our response to Comment 3.2), to minimize the confounding influence of families with narrow environmental distributions on broad taxonomic comparisons.

Regarding the reviewer's point on Fabaceae appearing to be uniquely dry-adapted, we would like to clarify that Fabaceae species in our dataset are broadly distributed across the full Amazon precipitation gradient (SM Fig 3). As seen in that figure, Fabaceae is not restricted to dry or highly seasonal forests but occurs widely, including in ever-wet regions. To further address this concern, we conducted ANOVA tests within each forest type, comparing Ψ_{50} values between Fabaceae and non-Fabaceae species (SM Fig 5). These analyses revealed no significant differences in Ψ_{50} within forests experiencing long dry seasons, supporting the idea that Fabaceae's high embolism resistance is not merely an artifact of excluding dry-adapted specialists with narrow ranges. These additional analyses and their interpretations are now presented in the Supplementary Material and referenced accordingly in the main text (Lines 308–314).

Comment 3.5. The authors also note that they exclude specific plots based on forest type and composition (L 503). But yet the resulting map is projected across the entire Amazon basin, without regard to forest type or composition. These decisions seem at odds with one another. Are the excluded plots regionally clustered or located on specific environmental conditions? If regions are excluded, then these should be masked from the extrapolation (or, more

rigorously, something like Multivariate Environmental Similarity Surfaces should be used to prevent environmental extrapolation).

Response: We appreciate the referee's thoughtful comment. We would like to clarify that the interpolation used to produce Figure 3 was employed as a visualization device only, to emphasize the broader, large scale pattern of Ψ_{50} in the basin. That is why we don't even include numeric values on the plot legend for the interpolated values. We fully agree that if the goal was to produce a high-resolution, spatially explicit model or 'layer' of embolism resistance across the Amazon, a more rigorous approach would be required, particularly to avoid extrapolation into unsampled environmental space. In that case, incorporating tools such as Multivariate Environmental Similarity Surfaces (MESS) or other environmental masking approaches would be essential. But while generating such a data layer would be an interesting and relevant research problem (which has been partially addressed by the recent publication by Chen, S. et al. *Nature* 631, 111–117 (2024). <https://doi.org/10.1038/s41586-024-07568-w> using satellite and environmental data to predict hydraulic safety margins), we feel it is beyond the scope of this paper, especially since, as recognized by the reviewer, the spatial distribution of the points is far from ideal to support quantitative modelling.

Masking forest type and composition is also a good suggestion. In response to it, we have improved our treatment of the interpolation by applying a mask to restrict the visualization to *terra-firme* forests (Revised Fig 3, which is also shown below). Specifically, we now exclude flooded forests, white-sand forests, and deforested areas, which differ substantially in edaphic and ecological conditions and were not included in our core analysis. These areas were masked using data from MapBiomas Collection 8 and the wetlands dataset by Hess et al. (2003). We believe these adjustments help ensure that the visualization aligns better with the data used in the analysis.

Fig 5. (Revised Figure 3). Estimated basin-wide spatial variation of Amazonian vulnerability to embolism. A. Macroecological patterns of community weighted mean value of Ψ_{50} , created using Inverse Weighted Distance interpolation of 448 upland moist forest plots distributed across Amazonia *sensu latissimo* (see methods). Our analyses exclude dry forests, flooded

forests, and plots with elevation >1000 m above sea level, as well as those affected by direct human disturbance. B. Estimated community weighted mean values for all 448 non-flooded forest plots across the basin. Each dot represents a forest plot and the colour shows its community-weighted mean Ψ_{50} . Due to the spatial scale of the map, plot locations in panel B are displaced (jittered) to remove overlap and improve visualization; exact plot locations are shown in panel. On both panels, we masked out areas of Amazonia with very different environments, notably flooded forests, white sand forests and deforested areas, to emphasise our focus on upland (*terra-firme*) moist forests.

Comment 3.6. In line with this, the location of the forest plots is very highly clustered (Fig. 3), and there's no attempt to quantify the relative extent of spatial extrapolation (or environmental extrapolation). In such a setting, using inverse distance to interpolate the space seems suspect, and at a minimum, I would expect to see how predictive accuracy decreases as a function of decay distance. There's no reason to expect that regions 500 km from observed data points can be interpolated accurately, and the resulting map (Fig 3a) supports this, as it essentially just outlines the strong clusters with no ability to truly interpolate the space between.

Response: The reviewer is correct in stating that the absolute accuracy of IDW interpolation would not be high in this case. However, we emphasize that our use of IDW was solely as a visualization tool to illustrate broader basin-wide patterns and highlight spatial clustering, rather than for precise estimations at unsampled locations. This approach has been previously applied in ecological and remote sensing studies, such as Mitchard et al. (2014), who used IDW to generate a biomass map from forestplots.net clusters. Their study demonstrated the utility of IDW in capturing broad-scale spatial trends from irregularly distributed field data while acknowledging its limitations for precise spatial predictions. Similarly, our application of IDW serves to outline the major clusters in space rather than to provide absolute values with high local accuracy. See our response below for a further exploration of the dataset to accommodate this.

Reference: Mitchard, E.T., Feldpausch, T.R., Brienen, R.J., Lopez-Gonzalez, G., Monteagudo, A., Baker, T.R., Lewis, S.L., Lloyd, J., Quesada, C.A., Gloor, M. and Ter Steege, H., 2014. Markedly divergent estimates of Amazon forest carbon density from ground plots and satellites. *Global ecology and biogeography*, 23(8), pp.935-946.

Comment 3.7. To obtain a realistic goodness of fit for the mapping results, I'd like see how these models perform using more rigorous spatially buffered leave-one-out CV or similar, using a kernel distance at which spatial autocorrelation drops to near zero (see Roberts et al. 2017, doi:10.1111/ecog.02881; Ploton et al 2020, doi: 10.1038/s41467-020-18321-y). I would also like to see the full raw cross-validation predictions for all LOO points, rather than the R2 of means (as in Fig. S5, which surely overestimates R2). I'd also be interested in seeing this cross-validation accuracy split up between the non-gap-filled points (L 508) and the gap-filled points, as I suspect the model performs worse if only worse due to the gap filling. I was also surprised not to see any spatial uncertainty, e.g., at a minimum using resampling or bootstrap approaches to quantify the role of sampling noise. As such, it's difficult to interpret the map with any confidence or make reliable inferences, apart from what is given in the raw plot data (e.g., Fig. 3B).

Response: To accurately estimate spatial autocorrelation (i.e. a semivariogram) we would need a uniform distribution of the distances among points, from near to very far. With our clustered point distribution, no acceptable semivariance model could be fit, a problem also encountered by Mitchard et al. (2014), who used IDW to generate a biomass map from forestplots.net clusters. What we opted to do instead was to apply a spatially-constrained environmental clustering algorithm to the dataset (using the ClustGeo R package, <https://cran.r-project.org/web/packages/ClustGeo/index.html>). Using this algorithm, we first clustered the data points into spatially coherent groups of similar Ψ_{50} , MCWD and Water Table Depth values. We then used a k-fold validation strategy excluding each cluster in turn to generate 5 separate IDW interpolations, quantifying the errors using the remaining cluster data. This ensures that the validation data is not spatially correlated with the training data at each fold. Our estimated RMSE averaged over the 5 folds using this strategy was 0.24 (s.d. 0.05), and we included the range of RMSE values in the main text on lines 596-599. To further smooth out the final visualization and better incorporate the error variability, we produced a final interpolation map by averaging the five interpolated maps resulting from each fold. We also added SM Fig 7 (presented below) into the manuscript to show the results of the clustering, to further emphasize the focus on the broader geographic pattern. Finally, we modified the text to make it more clear to users that the resulting 'map' should not be considered an accurate representation of absolute Ψ_{50} values over the Amazon, but as a visualization of a broad geographic pattern. In fact, we removed all mentions of the word map and refer to it instead as a macroecological/geographic pattern.

Fig. 6 (New SM Figure 7). Top: results of the geographically constrained clustering of plots based on the combination of Ψ_{50} , MCWD (Mean Cumulative Water Deficit) and WTD (Water Table Depth) for the 448 inventoried non-flooded upland moist forest plots distributed across Amazonia *sensu latissimo* (see methods). Areas with very different edaphic environments - flooded forests, white sand forests and deforested areas – are masked from the background. Bottom: Distribution of Ψ_{50} values within each spatial cluster.

Comment 3.8. Lastly, given that the authors wish to explore the role of environmental drivers like water depth, is there a reason they did not use other mapping/modelling approaches that explicitly link embolism to covariates? The current approach of a simple linear model (Fig 8C) is likely confounded by a variety of underlying abiotic factors, such as precipitation and temperature regimes, soil types, etc. If the authors want to make robust inferences about water table depth, I would suggest a more structured approach to better control for confounders and interactions.

Response: We agree with the reviewer, but as answered above for comment 3.5, our intent was not to quantitatively and accurately predict Ψ_{50} values spatially across the basin nor to generate a 'data layer', but rather to demonstrate the existence of a broad macroecological/geographic pattern. This level of complex modelling is worth investigating in the future (and in fact has been already approached, as we mentioned, in terms of hydraulic safety margin by Chen et al. (2024)), but this would comprise an entirely different study and manuscript.

Nevertheless, we would like to emphasize the fundamental role of our expanded community-level dataset (achieved through phylogenetic imputation) in, for example, supporting remote sensing efforts to assess forest resistance and resilience from space. Remotely sensed vegetation indices have been widely used to investigate resilience loss in forests (e.g. Forzieri et al., 2022; Tai et al., 2022), but ground-based validation of these approaches remains scarce. Our study provides crucial empirical insights into vegetation resistance to embolism, based on an extensive dataset spanning 448 forest sites across Amazonia. This large-scale dataset can be harnessed to validate and improve future remote-sensing based models quantifying forest resilience, and to enhance the interpretation of remote sensing indicators, thus improving our ability to detect early warning signals of forest functioning decline.

Reference:

Chen, S., Stark, S.C., Nobre, A.D., Cuartas, L.A., de Jesus Amore, D., Restrepo-Coupe, N., Smith, M.N., Chitra-Tarak, R., Ko, H., Nelson, B.W. and Saleska, S.R., 2024. Amazon forest biogeography predicts resilience and vulnerability to drought. *Nature*, 631(8019), pp.111-117.

Tai, X., Trugman, A.T. and Anderegg, W.R., 2023. Linking remotely sensed ecosystem resilience with forest mortality across the continental United States. *Global Change Biology*, 29(4), pp.1096-1105.

Forzieri, G., Dakos, V., McDowell, N.G., Ramdane, A. and Cescatti, A., 2022. Emerging signals of declining forest resilience under climate change. *Nature*, 608(7923), pp.534-539.

A few minor comments:

Comment 3.9. Supplementary Table 1 appears to be missing

Response: We thank the referee for spotting this mistake. Supplementary Table 1 refers to an analysis that was part of an earlier version of the manuscript but has since been removed during revisions. We have now updated the supplementary materials to reflect the current version of the manuscript and removed the reference to this table to avoid confusion.

Comment 3.10. What dataset did the water table depth come from? Is this Mattos et al?

Response: We thank the referee for spotting this mistake. Information about water table depth was obtained from a raster file available at Fan et al., (2013). We added this information in the legend of SM Fig 8 and methods.

Reference: Fan, Y., Li, H. and Miguez-Macho, G., 2013. Global patterns of groundwater table depth. *Science*, 339(6122), pp.940-943.

<http://thredds-gfnl.usc.es/thredds/catalog/GLOBALWTDFTP/catalog.html>

Comment 3.11. The data description could have a bit more detail. Are all of the analyses and plots species-level averages? Did the original dataset contain multiple measurements within each species? If so, what the the level of intra vs. interspecific variation in the full dataset?

Response: We have now added a more detailed description of the dataset and analytical structure to the Methods section (please, see lines 467–470). The Ψ_{50} data available from Tavares et al., (2023) consists of one fitted vulnerability curve per species per plot, based on pooled data from individuals of the same species at each site. Because of this structure, our dataset does not include multiple individual-level measurements within species at the same site, which limits our ability to explicitly quantify intra- versus interspecific variation in embolism resistance.

Responses to referees
Manuscript “Family imprint reveals basin-wide patterns of Amazon forest embolism
resistance”

Tavares et al.

Tracking Number NCOMMS-24-52403A

Referees' comments:

Reviewer #1 (Remarks to the Author):

Comment 1.1. The MS first claims to show strong phylogenetic signal in xylem vulnerability, which is then used to impute Ψ_{50} in forest plots where species were not measured, then interpolated across the landscape to create a “resistance to embolism?” map. This is an interesting method that creates a nice overview of possible landscape function, but the accuracy of this approach is highly questionable in my opinion.

Response: We thank the referee for highlighting the importance of the study and we have tried to clarify the referee’s concerns over the uncertainty of our approach in the next comments.

Comment 1.2. A key finding is stated as the demonstration that Fabaceae are more embolism resistant than other families in Amazonia. Combined with the finding that “family ($F=2.74$, $p=0.02$) is a stronger predictor of Ψ_{50} than genus ($F=1.88$, $p=0.07$) or species ($F=1.1$, $p=0.4$)”. This leads the reader to assume that much of the imputation is driven by family-level means. The problem with this approach is that the variation in Ψ_{50} in Fabaceae spanned almost the entire range of all other taxa. When combined with the fact that the Fabaceae mean Ψ_{50} was not significantly different from other families, one begins to see that the (potentially adaptive) variation that is likely to be captured by the predictions is extremely low.

Response: We agree with the referee that Fabaceae exhibits a wide range of Ψ_{50} values in our dataset. However, it is important to note that Fabaceae species are broadly distributed across the entire Amazon precipitation gradient (SM Fig. 3), including ever-wet and intermediate and long DSL regions, and consistently rank among the more embolism-resistant taxa within each site (Fig. 1; now also added to the SM Fig. 6). This pattern indicates that, despite local environmental variation, Fabaceae tend to occupy the resistant end of the hydraulic spectrum across Amazonia forest plots (lines 310-313).

Because of this within-family variation, mean Ψ_{50} values do not differ significantly when comparing Fabaceae against other families with wide occurrences (those present in ≥ 3 sites; Manuscript Fig. 2A; see lines 289–297 and 518–522). Nevertheless, when Fabaceae are compared to all non-Fabaceae taxa combined, they emerge as markedly more embolism resistant (Manuscript Fig. 2B). This underscores that while the family spans a diversity of hydraulic strategies, Fabaceae as a whole tends to consistently be represented among the most resistant species in Amazonian forests.

Fig. 1 (New SM Figure 6) | Embolism resistance variation across sites (Tavares et al., 2023), which are ordered left to right as drier (NVX) to wetter (SUC) environments: long dry season length - DSL (NVX, KEN1, KEN2); intermediate DSL (FEC, TAP, CAX, TAM, MAN) and ever-wet aseasonal forests (ALP1, ALP2, SUC). Boxplots show the 25th percentile, median and 75th percentile. Vertical bars show the interquartile range $\times 1.5$ and data points beyond these bars are potential outliers. The dashed horizontal line shows the mean Ψ_{50} across the pan-Amazonian dataset. Red points represent Fabaceae species.

Comment 1.3. Of the 3 outputs of this paper, none seem very robust.

1. That phylogeny has an influence on P50 is already known
2. That Fabaceae is particularly tough does not seem particularly well supported considering the enormous diversity of P50 in the family (presumably correlated with the large diversity of the family).
3. That coarse scale mapping on P50 provides a useful insight into regional-scale “vulnerability” seems unlikely or at least demanding of a far more robust ground-truthing.

Response: We appreciate the referee’s comments and the opportunity to clarify the major contributions of our study. While previous global studies have demonstrated phylogenetic conservatism in Ψ_{50} (e.g: Laughlin et al. 2023; Sanchez-Martinez et al. 2023), these analyses included extremely limited representation of Amazonian taxa (please see figure below).

Fig. 2 | Geographic distribution of plots used to evaluate in global assessments of embolism resistance phylogenetic conservatism. Images from (Laughlin et al. 2023; Sanchez-Martinez et al. 2023).

Our work fills this gap by showing that embolism resistance is also phylogenetically conserved *within Amazonia*, across 129 species spanning the full precipitation gradient. Phylogenetic signal often weakens at regional scales compared to global analyses (Swenson et al. 2006; Krasnov, Poulin, and Mouillot 2011; Graham et al. 2018; Losos 2008), so demonstrating this pattern within a single biome adds important evidence for the evolutionary structuring of hydraulic traits in the largest and most diverse tropical forest of the world.

We agree that Fabaceae spans a broad range of Ψ_{50} values, reflecting the environmental and taxonomic diversity of the family. However, a key finding is that Fabaceae species consistently occupy the resistant end of the spectrum within sites across the entire precipitation gradient (see response to Comment 1.2 and Fig. 1). Thus, despite the within-family variation, Fabaceae are disproportionately represented among the most embolism-resistant species within Amazonian forest plots, making their hydraulic strategy ecologically significant.

We agree that the IDW-based map should not be interpreted as a predictive “data layer”, but rather as a visual guide to understand large-scale biogeographical patterns. Please see the detailed responses given to referee 3 below.

References:

- Graham, Catherine H., David Storch, Antonin Machac, and Nick Isaac. 2018. “Phylogenetic Scale in Ecology and Evolution.” *Global Ecology and Biogeography: A Journal of Macroecology* 27 (2): 175–87.
- Krasnov, Boris R., Robert Poulin, and David Mouillot. 2011. “Scale-Dependence of Phylogenetic Signal in Ecological Traits of Ectoparasites.” *Ecography* 34 (1): 114–22.
- Laughlin, Daniel C., Andrew Siefert, Jesse R. Fleri, Shersingh Joseph Tumber-d, William M.

- Hammond, Francesco Maria Sabatini, Gabriella Damasceno, et al. 2023. "Rooting Depth and Xylem Vulnerability Are Independent Woody Plant Traits Jointly Selected by Aridity, Seasonality, and Water Table Depth." <https://doi.org/10.1111/nph.19276>.
- Losos, Jonathan B. 2008. "Phylogenetic Niche Conservatism, Phylogenetic Signal and the Relationship between Phylogenetic Relatedness and Ecological Similarity among Species." *Ecology Letters* 11 (10): 995–1003.
- Sanchez-Martinez, Pablo, Maurizio Mencuccini, Raúl García-Valdés, William M. Hammond, Josep M. Serra-Diaz, Wen Yong Guo, Ricardo A. Segovia, et al. 2023. "Increased Hydraulic Risk in Assemblages of Woody Plant Species Predicts Spatial Patterns of Drought-Induced Mortality." *Nature Ecology & Evolution*. <https://doi.org/10.1038/s41559-023-02180-z>.
- Swenson, Nathan G., Brian J. Enquist, Jason Pither, Jill Thompson, and Jess K. Zimmerman. 2006. "The Problem and Promise of Scale Dependency in Community Phylogenetics." *Ecology* 87 (10): 2418–24.

Comment 1.4. Thus, I feel that a much more cautious approach to the interpretation of these data is warranted. Having said this, I do believe there is promise in such an approach, so perhaps the manuscript simply requires a more circumspect rather than categorical viewpoint. Response: We thank the referee for this thoughtful comment. We agree that a cautious interpretation is necessary given the inherent uncertainties of trait extrapolation and spatial interpolation, and we have revised the manuscript to adopt a more circumspect tone throughout in the Discussion (lines, 426-428, 431-433, 452-459). Specifically, we now emphasise that our findings highlight patterns and hypotheses emerging from the largest Amazon-wide hydraulic trait dataset to date, rather than categorical statements about the absolute values of embolism resistance across the basin. At the same time, we believe that our work demonstrates the promise of combining phylogenetic imputation with community inventory data to explore hydraulic strategies at macroecological scales. By grounding the analysis in 129 species across 11 sites spanning the full Amazonian precipitation gradient, and explicitly discussing the limitations of our approach (lines 581-582, 600-612), we aim to provide a robust empirical framework that can guide future data collection and modelling efforts.

Reviewer #2 (Remarks to the Author):

Comment 2.1. Dear Authors,

I found authors revised the paper throughout in response to reviewers' comments and questions inclusive of my own.

Added information in review responses added further information. Revised paper presents clear and new insights from the region-wide phylogenetic and geographic analysis focusing on stem embolism tolerance are worth to be published. Reviewer 2

Response: We thank the referee for their positive assessment and encouraging feedback. We are pleased that the revisions and additional information clarified our approach. We greatly appreciate the constructive comments provided throughout the review process, which have significantly strengthened the manuscript.

Reviewer #3 (Remarks to the Author):

Comment 3.1. I appreciate the clarification the authors have provided regarding the subsetting of data and the comparison of Fabaceae relative to other families. The additional analysis and clarification of the text address my key concerns in this area, providing convincing evidence that Fabaceae is not simply a uniquely widespread drought-tolerant specialist. I likewise appreciate the exclusion of specific forest types in the spatial projection, as well as the clustered cross-validation for overall model fit. Though, for this last point, what is the R^2 of observed vs. predicted rather than the RMSE? Depending on how the data are normalised, an RMSE of 0.24 could equate to an R^2 as low as $RMSE^2 = 0.05$ (see Li 2017 for a discussion on this point <https://doi.org/10.1371/journal.pone.0183250>).

Response: Thank you for your comments. We do not understand however why the reviewer is asking to see the R^2 values, when the discussion in the suggested reference clearly states that r and r^2 are not appropriate measures of prediction accuracy when comparing observed vs predicted values. As the authors state in the suggested reference: "The r and r^2 do not measure the accuracy and are incorrect accuracy measures. The existing error measures suffer limitations. VE_{cv} and $E1$ are recommended for assessing the accuracy. The applications of these accuracy measures would encourage accuracy-improved predictive models to be developed to generate predictions for evidence-informed decision-making". We have thus calculated VE_{cv} (Variance Explained) values as per the recommendation of the suggested paper, and present these results below.

Comment 3.2. With this in mind, there is still inadequate assessment of the uncertainty in the spatial projections. The authors state that they are unable to do traditional spatial uncertainty/extrapolation analyses because "the spatial distribution of the points is far from ideal to support quantitative modelling," and that, "with our clustered point distribution, no acceptable semivariance model could be fit." But this is exactly the issue -- if the data do not allow for appropriate statistical analysis, then how is the analysis justified?

Comment 3.3. The authors state that, "our intent was not to quantitatively and accurately predict Ψ_{50} values spatially across the basin..." and that determining the extent of spatial extrapolation is, "beyond the scope of this paper..." Yet they are visually depicting these spatial patterns and referencing these patterns in the title, so one would hope that doing so "accurately" is an aim.

Response to 3.2 and 3.3: We apologise if our statement was not clear before. Our data is clustered spatially, which prevents the use of the specific statistical method of kriging, as suggested before. Other methods of quantitative modelling (e.g. regression, random forests or other types of continuous predictive modelling based on other explanatory variables will be further limited by the mismatch between the spatial resolution of environmental layers (often 1km pixels or more) and the ecological scale that would be relevant for such modelling. Tree physiology is strongly determined by microclimate and microtopography, usually varying on a scale of a few meters to a few hundreds of meters (<https://doi.org/10.1088/1748->

9326/ad0064). This local scale variability is not well captured by existing environmental layers, which justifies why such modelling approaches are yet to be used successfully for predicting Ψ_{50} .

However, we are not attempting to generate accurate spatial predictions of Ψ_{50} for every mapped location/pixel (i.e. a Ψ_{50} ‘data layer’). We are using mathematical (not statistical) spatial interpolation methods to further visualise the broad biogeographical patterns in the distribution of Ψ_{50} across the Amazon basin, already shown by the data. This is why we have opted to use Inverse Distance Weighted interpolation, a simpler and recognised mathematical method of spatial interpolation that makes no statistical assumptions, unlike kriging. IDW aids in *visualisation* of the general spatial patterns as we intend. And for this very reason, we have refrained from showing actual Ψ_{50} values on the legend of Figure 3A, and only provide actual values on Figure 3B (which shows actual plot-level values, not interpolated values).

There is a large difference between predicting Ψ_{50} values accurately at each location (i.e. creating a ‘data layer’) and using spatial interpolation methods to visualise broad spatial patterns and regions with higher and lower Ψ_{50} values. For example, Bergmann’s rule is a very well known and recognised biogeographical spatial pattern where endotherms tend to have larger body mass towards the poles and less body mass towards the tropics. If we interpolated point data on body mass across the globe using IDW we would be able to visualise this pattern - yet no one would expect Bergmann’s rule to accurately predict body size at a given location for any taxa (<https://doi.org/10.1111/gcb.16860>). In the same vein, all we are affirming in our manuscript is that some regions in the Amazon tend to have larger Ψ_{50} values than others - and IDW helps us better visualise these patterns than plotting actual values or the clustering results alone - but we cannot with our data predict actual Ψ_{50} values with enough accuracy to confidently present them as a ‘spatial data layer’. Still, we strongly believe evidencing these broad patterns provides a valuable contribution to understanding how biogeography plays a role in the vulnerability of Amazonian forests to more frequent and stronger droughts. We have further emphasised the nature of the analysis as a visualisation only on lines 581-582 and 610-612.

With that in mind, we have performed additional uncertainty assessment as requested by the reviewer and emphasised by the associate editor, but we would ask you to please keep the above in mind as we reply to the comments. It is expected that validation scores based on point-based observed vs ‘predicted’ (not really a prediction, as stated above) scores will be low - especially as we spatially smooth the IDW results for better *visualization*. therefore reducing the range of Ψ_{50} in the interpolated map. However, we also provide further evidence that the visual pattern of Ψ_{50} distribution is correct (see answer below).

Comment 3.4. These calculations (the MESS statistics, spatial uncertainty via bootstrapping, etc) are relatively easy to do with so few points, and would provide important context for understanding which regions are being estimated with higher/lower uncertainty.

Response: In the previous review we assessed uncertainty via k-fold cross-validation using environmental-spatial clustering to define the folds for training and test samples. We used this strategy instead of bootstrapping as it controls for the spatial autocorrelation between training and test data - at each fold the validation data is from a different geographical region than the training data, ensuring independence between training and testing. For this review, we included

the calculation of Variance Explained (VEcv) as per the reference suggested by the reviewer. We have also calculated the MESS statistic as requested, using the MCWD and Water Table Depth layers as environmental variables and extracting their corresponding values for each of the 448 plots as the reference values for MESS calculation. We have modified the text on lines 600 to 638 to further describe these analysis and the validation results, and included three additional figures as Supplementary Material to further support it.

Overall, the the IDW interpolation reproduces the patterns already revealed by clustering, with more negative Ψ_{50} values for the northern and southeastern Amazon. This can be seen on Supplementary Material Figure 7, reproduced below:

Fig. 3 (SM Figure 8) | Top: results of the geographically constrained clustering of plots based on the combination of Ψ_{50} , MCWD (Mean Cumulative Water Deficit) and WTD (Water Table Depth) for the 448 inventoried non-flooded upland moist forest plots distributed across Amazonia sensu latissimo (see methods). Areas with very different edaphic environments - flooded forests, white sand forests and deforested areas – are masked from the background. Bottom: Distribution of Ψ_{50} values within each spatial cluster..

The uncertainty analysis using k-fold cross validation, as expected, does not show a very strong agreement with observed data (for the reasons explained in the response to comments 3.2 and 3.3), as seen on the new Supplementary Material Figure 12, below:

Fig. 4 (New SM Figure 13) | Observed versus interpolated (predicted) Ψ_{50} values based on five-fold cross-validation, showing how the smoothing and averaging that results from the IDW interpolation method used reduced the range of Ψ_{50} values to between -2 MPa and -2.4 MPa, versus -1.3 MPa to -3.8 MPa in the observed data, and explaining the low quantitative agreement at the local (point/pixel) level.

Root Mean Square Error for the k-fold crossvalidation was of 0.24 MPa, with the value for each fold varying from 0.19 MPa to 0.32 MPa. We can see from the plot above that most of the mismatch between observed and ‘predicted’ values comes from the smoothing of ‘predicted’ values that is part of the IDW interpolation. While observed data ranged from -1.3 MPa to -3.8 MPa, the ‘predicted’ data only ranged from -2 MPa to -2.4 MPa. For this reason, the calculated Variance Explained (VEcv) was only 6.7%.

While this value would be considered very low for a quantitative predictive model, we emphasize again that our use of IDW is just to provide a better and more continuous visual

representation of the broad scale Ψ_{50} patterns already revealed by clustering. To demonstrate that, we have first calculated the mean Ψ_{50} value for each spatial cluster using the source point data (*point averages*). We then delineated the minimum convex hull enclosing all data points in each cluster, and used these polygons to extract all Ψ_{50} values from the final interpolated raster, which were averaged to provide an second mean Ψ_{50} value calculated from the interpolated values (*interpolated averages*). If the IDW interpolation is indeed reproducing the general spatial pattern already shown by the clustering, we would expect these two averages to agree well. And this is what we in fact observe (new Figure SM 13, reproduced below).

Fig. 5 (New SM Figure 14) | Agreement between average Ψ_{50} values calculated from the field-based estimates, for each spatial cluster (SM Figure 7), and average Ψ_{50} values calculated from interpolated data, for the area determined by the minimum convex hull enclosing each spatial cluster.

The MESS analysis did not reveal any strong environmental disagreement between the reference locations and the full interpolation domain (*Amazonia latissimo sensu*). Other than isolated pixels, only the highland Andean region and a few regions in lowland western Amazonia were outside of the range of the reference data (i.e. MESS values below 0 or above 100, new Figure SM 14, reproduced below). Overall, 93% of the entire area had MESS values

within the 0 to 100 range, further indicating environmental conditions were mostly within the reference range. MESS values ranged from -846.5 to 95.7, with a median of 13.6 and an average of 15.6.

Fig. 5 (New SM Figure 15) | Multivariate Environmental Similarity Surfaces (MESS) analysis of the interpolated surface, using Maximum Cumulative Water Depth and Water Table Depth layers as environmental variables, and their corresponding values for each of the 448 plots as the reference values for MESS calculation.

Comment 3.5. The justification the authors provide for their approach is Mitchard et al. (2014). But in the decade since this paper was published, there has been growing consensus about the appropriate analysis of spatial patterns and what is necessary when producing spatial projections. It's certainly possible that the spatial projections are reasonable, but without estimates of uncertainty/extrapolation, it's equally likely that the patterns are heavily biased. The uncertainty/extrapolation analysis doesn't need to be incredibly laborious, but even some indication of which regions have higher/lower uncertainty would help. If the data do not allow for proper error assessment, or if the spatial projections have a small area of applicability or low goodness of fit, then I'm not sure that a spatial analysis is feasible given the current data.

Response: Please see our responses above clarifying the nature of our analysis and the expanded analysis of uncertainty / error assessment as requested.

Responses to referees
Manuscript “Family imprint reveals basin-wide patterns of
Amazon forest embolism resistance”

Tavares et al.

Tracking Number NCOMMS-24-52403B

Reviewers Comments

Reviewer #1 (Remarks to the Author):

Comment 1.1. Authors have responded to my previous comments with some justifications in their rebuttal, but without making any substantive changes to the text. My suggestion that there should be more consideration of potential weaknesses in the proposed arguments, has been ignored in the text. This is OK I suppose, but it does seem unfortunate to make bold statements about climate vulnerability, when it is possible that P50 patterns may more strongly reflect phylogenetic patterns than tolerance of drought, Consider the case if Fabaceae tend to be more shallow rooted than other families, their more negative P50 will not correspond to a greater capacity to withstand drought, but rather a compensation for their shallow roots. Furthermore, if the presence of Fabaceae exerts a major influence on community “vulnerability”, then this may not reflect a sensitivity of the community to rainfall shortage in the way that is being sold here. This is a theoretical scenario, but it is based on classical observations in the field, and by the general patterns observed in Choat et al 2012 (Nature).

Response: We thank the referee for this thoughtful comment and for highlighting the importance of carefully distinguishing between different mechanisms underlying drought response. We agree that a more cautious interpretation is warranted, and we have revised the manuscript to adopt a less categorical tone throughout the Results and Discussion.

Regarding the referee's theoretical scenario: “*if Fabaceae tend to be more shallow rooted than other families, their more negative Ψ_{50} will not correspond to a greater capacity to withstand drought, but rather a compensation for their shallow roots*” - we respectfully suggest that this scenario represents a drought avoidance vs. tolerance distinction rather than a contradiction of our interpretation (Santiago et al., 2018; Laughlin et al., 2023). Deep-rooted species rely on drought avoidance, while shallow-rooted species likely require more embolism-resistant xylem to tolerate low water potentials (Brum et al., 2018; Oliveira et al., 2021). These strategies are not mutually exclusive and do not undermine the ecological meaning of lower Ψ_{50} values, instead, they highlight that hydraulic traits must be interpreted within a broader whole-plant framework. That said, we fully agree with the referee that Ψ_{50} alone cannot capture the full complexity of drought resistance (Choat et al., 2018). In response, we have now strengthened the manuscript by adding explicit caveats about the limitations of interpreting Ψ_{50} as a direct proxy for drought vulnerability and clarifying that our conclusions concern basin-wide patterns of embolism resistance, not absolute drought resistance across all possible mechanisms.

We agree that community-level patterns of Ψ_{50} may partially reflect phylogenetic composition. However, this does not preclude ecological interpretation, instead, it emphasises the central role of evolutionary history in shaping hydraulic strategies. To make this clear, we now explicitly state that our spatial patterns represent embolism resistance distributions, not direct predictions of drought mortality.

Finally, we appreciate the referee's recognition that our approach holds promise, as noted in their previous round of comments. In the revised manuscript, we now explicitly emphasise that our results identify emergent macroecological patterns and hypotheses, rather than providing definitive predictions of climate-change vulnerability. We agree that adopting a more circumspect tone strengthens the paper. By grounding our analysis in an Amazon-wide empirical dataset and openly acknowledging its limitations, we aim to offer a robust foundation for future data collection and modelling efforts, as well as for integrative research linking ecophysiology, demography, and forest dynamics. Please, the changes in the revised manuscript are marked as yellow.

References:

Santiago, Louis S., et al. "Coordination and trade-offs among hydraulic safety, efficiency and drought avoidance traits in Amazonian rainforest canopy tree species." *New Phytologist* 218.3 (2018): 1015-1024.

Laughlin, Daniel C., et al. "Rooting depth and xylem vulnerability are independent woody plant traits jointly selected by aridity, seasonality, and water table depth." *New Phytologist* 240.5 (2023): 1774-1787.

Brum, Mauro, et al. "Hydrological niche segregation defines forest structure and drought tolerance strategies in a seasonal Amazon forest." *Journal of Ecology* 107.1 (2019): 318-333.

Oliveira, Rafael S., et al. "Linking plant hydraulics and the fast-slow continuum to understand resilience to drought in tropical ecosystems." *New Phytologist* 230.3 (2021): 904-923.

Choat, Brendan, et al. "Triggers of tree mortality under drought." *Nature* 558.7711 (2018): 531-539.

Reviewer #3 (Remarks to the Author):

Comment 2.1. I appreciate the substantial effort the authors have invested in the additional spatial modelling and the clarity of analysis. I have no critiques with the overall question or analysis, nor do I have an issue with showing a spatial map per se, but I continue to remain skeptical about the robustness and usefulness of the current mapping approach.

Response: We thank the referee and acknowledge their personal position. We feel that after addressing their concerns and including the additional analyses requested, the manuscript now offers enough additional information and support to allow readers to reach their own conclusions (as requested below) about our general spatial patterns. Moreover, we added additional caveats regarding the analysis and figure in three different sections of the manuscript, in response to editor requests - please see the attached Author Checklist and the changes in the revised manuscript, which are marked as yellow.

Comment 2.2. As stated previously, my main concern is extrapolation beyond data support. Large parts of the central basin lack observations; thus the maps necessarily predict into unsampled regions. Despite the authors' reframing this as simply a "visualization", this is misleading---it is still an extrapolation into regions where there are no data---and needs to be

evaluated as such. Ultimately, we need a way to distinguish reliable spatial projections from poor ones, or at least provide the readers with the necessary information for them to judge on their own.

Response: We understand the referee position but still disagree fundamentally with the perception that there is no distinction between visualising/spatializing general patterns and providing accurate and fully quantitative spatial predictions. We agree that our work extrapolates into regions where there are no data - this is the fundamental aspect of the paper. But we believe at this point we have made it clear that while we are confident we are capturing broad biogeographic/macroecological patterns, there is still much work to be done in further confirming and 'unpacking' these patterns. We have refrained since the first version of the paper to make any quantitative claims or support any quantitative inference from our spatialization, exactly because we agree with the referee that it could be misleading - and it is in fact why although we report the code used to generate the data in the figure (as required by the journal), we are not making the actual results available as a 'data layer'. It does not have the required level of quantitative accuracy we would deem necessary to release it as such. But the visualization and discussion of the broad biogeographic patterns is still valid, and very relevant and important to drive forward the discussion about the vulnerability and the future of the Amazon biome. As stated by the referee, we are confident we now provide all the necessary information for readers to judge on their own.

Comment 2.3. Regarding predictive performance, I fully agree with the authors that spatial projections need not (and rarely do) deliver precise point estimates, but they must be approximately unbiased (i.e., track the 1:1 line) even if they have high point-to-point variance. That is, the discussion invoking Bergmann's rule appears to conflate inference about the mean trend with the width of prediction intervals; wide prediction intervals (low predictive accuracy at the point level) do not justify biased trends on aggregate. The new cross-validation (SM Fig. 13) suggests negligible predictive accuracy with substantial bias. And while SM Fig. 14 indicates good within-fit performance on aggregate, to justify extrapolating trends beyond clustered regions, this likewise needs to be conducted out-of-fit. However, even if the authors were to aggregate the clusters in SM Fig. 13 and plot obs vs. pred for each of the 5 folds, it appears that there would be no real signal, suggesting that even on aggregate, the Kriging approach does not yield approximately unbiased results.

Response: First, we must reiterate we did not use a Kriging approach in this work, exactly because it requires a strong spatial autocorrelation model that is not present in our dataset due to geographic sampling bias. Our first version of this visualisation included only the interpolated data points based on distance. Since then, we introduced the clustering analysis to our method - addressing the referee's suggestion that the environment should be better represented. We also performed the MESS validation as requested to prove that although we are extrapolating to regions with no data, these regions are still within the environmental envelope covered under the point data. We have also already clarified (using Fig SM13) that most of the bias results from the inherent smoothing of the IDW method, which we further smooth to avoid unnatural sharp transitions in values. This bias would have been an issue if we were making any quantitative predictions from the data, but all we are showing is the very broad variations in the underlying averages (i.e. inferences about the mean as stated above). We disagree that there is no real signal, and we believe figure SM 14 clearly shows this - plotting the average observations per cluster versus the average interpolated values per cluster show a very close agreement. We thus hope that the multiple additions to the

method we have made during the review process are now sufficient for readers to make their own conclusions.

Comment 2.4. The MESS analysis is a useful direction, but the variable choice needs clarification. Much of the text emphasizes Mean Cumulative Water Deficit and Water Table Depth, whereas the MESS uses “Maximum Cumulative Water Depth.” If this is not a typo, please justify the choice and the exclusion of other plausible drivers (e.g., temperature, precipitation/BioClim, topography). Because MESS depends on the included variables, restricting to two requires a rationale; alternatively, show that these two suffice by demonstrating relatively unbiased spatial CV performance with them.

Response: First, we apologise for the typo. It should have said Maximum Cumulative Water Deficit on the Figure 15 SM legend and this has now been corrected.

We have chosen to use MCWD and WTD as our predictor variables because these are two of the variables with the most direct causal relation with plant water stress and causes of embolism. They encompass several basal climatic factors (temperature, rainfall, evapotranspiration, VPD), and we have shown previously (Tavares et al. 2023 Nature 617:11–117) that MCWD has a strong correlation to Ψ_{50} values. As highlighted in the previous round of review responses, detailed modelling of the environmental controls on Ψ_{50} would be a worthy scientific problem but is outside the scope and purpose of this paper. Again, our aim is to better understand spatial patterns in Ψ_{50} , and not to quantitatively estimate Ψ_{50} on a per-pixel basis across the Basin. We therefore believe we are using MESS as intended, to demonstrate that as far as WTD and MCWD are concerned, the plot locations from where data was interpolated (not statistically predicted) do cover the entire expected range of these variables in the Amazon. We feel at this point there is not much room left for further rebuttal, and that the referee and ourselves have very different perceptions and expectations - we could continue this debate for a long time. As stated by the referees themselves, the most important issue is that our analysis is transparent and well supported enough for its merit to be judged individually by readers, and we are confident we have reached this stage now after all the requested reviews.

Comment 2.5. Given the current CV results, I remain hesitant about including Fig. 3 as a basin-wide product. It remains very possible that it is either misleading or essentially random outside of the sampled areas. If the editor prefers to treat the figure as a qualitative visualization rather than a quantitatively accurate projection, I’m happy to defer to that framing. But in that case, I think more caveats would be required, and perhaps adding a much coarser categorical classification (high/med/low), which presumably can be predicted with greater accuracy.

Response: we appreciate the referee’s stance, but we disagree that it is misleading or essentially random outside of the sample areas. As the Ψ_{50} values approximated for the unsampled areas are entirely driven by a distance-decay function that averages across the existing samples - so by design correlated with the existing observations, it is not mathematically possible for these to be random. Moreover, we also have further confirmed the patterns using two independent methods: interpolation and clustering. We reiterate that we are not positioning Figure 3 as a data product of any kind, but rather presenting it as a visual aid to better show the patterns already present in the point data (which can be assessed by looking at Figure 3B, albeit not so immediately clear and readable - hence the improved visualization of Figure 3A). We have answered all previous and current comments

of the referee and have on their request modified our method and included several metrics of model performance and generality. We believe at this point there is sufficient information in the document and supplementary material for readers to come to their own conclusions about our results, and to revisit/rebut them with new data to further advance our knowledge on the vulnerability of Amazonian forests to ongoing climatic changes.